Corrected: Publisher correction; Author correction

# Tropical explosive volcanic eruptions can trigger El Niño by cooling tropical Africa

Myriam Khodri [1], Takeshi Izumo [1,2], Jérôme Vialard [1], Serge Janicot [1], Christophe Cassou[3], Matthieu Lengaigne [1,2], Juliette Mignot [1], Guillaume Gastineau [1], Eric Guilyardi [1,4], Nicolas Lebas[1], Alan Robock [5] & Michael J. McPhaden[6]

Stratospheric aerosols from large tropical explosive volcanic eruptions backscatter shortwave radiation and reduce the global mean surface temperature. Observations suggest that they also favour an El Niño within 2 years following the eruption. Modelling studies have, however, so far reached no consensus on either the sign or physical mechanism of El Niño response to volcanism. Here we show that an El Niño tends to peak during the year following large eruptions in simulations of the Fifth Coupled Model Intercomparison Project (CMIP5). Targeted climate model simulations further emphasize that Pinatubo-like eruptions tend to shorten La Niñas, lengthen El Niños and induce anomalous warming when occurring during neutral states. Volcanically induced cooling in tropical Africa weakens the West African monsoon, and the resulting atmospheric Kelvin wave drives equatorial westerly wind anomalies over the western Pacific. This wind anomaly is further amplified by air–sea interactions in the Pacific, favouring an El Niño-like response.

[1] Laboratoire d'Océanographie et du Climat: Expérimentations et approches numériques, Sorbonne Universités, UPMC Université Paris 06, IPSL, UMR CNRS/IRD/MNHN, F-75005 Paris, France. [2] Indo-French Cell for Water Sciences, IISc-NIO-IITM-IRD Joint International Laboratory, NIO, Goa 403002, India. [3] CECI, CNRS, Cerfacs, Université de Toulouse, Toulouse 31057, France. [4] NCAS-Climate, University of Reading, Reading RG6 6BB, UK. [5] Department of Environmental Sciences, Rutgers University, New Brunswick, NJ 08901, USA. [6] Pacific Marine Environmental Laboratory (PMEL), NOAA, Seattle, WA 98115, USA. Correspondence and requests for materials should be addressed to M.K. (email: myriam.khodri@locean-ipsl.upmc.fr)

The El Niño Southern Oscillation (ENSO) is the leading mode of natural interannual climate variability[1]. It is associated with large-scale sea surface temperature (SST) anomalies in the central and eastern Pacific. Those anomalies grow as the result of the Bjerknes positive feedback[2] between the ocean and atmosphere, whereby a central Pacific warm SST anomaly induces westerly wind anomalies that strengthen the initial SST anomaly. ENSO has two opposite phases with warm events being referred to as El Niño, and cold ones as La Niña. ENSO events occur spontaneously every couple of years, and usually peak at the end of the calendar year with worldwide impacts on climate through the alteration of the global atmospheric circulation[1].

Because of its global impacts and strong societal relevance, improving the understanding and forecasting of ENSO is a key objective in climate science. The understanding of ENSO intrinsic dynamics has improved over recent decades, in particular with the identification of useful predictors[3]. Equatorial Pacific heat content, measured as the volume of water above 20 °C equatorward of 5° in the Pacific (warm water volume (WWV)), is the most commonly used ENSO predictor[4]. An unusually high early-year WWV tends to precede an El Niño peaking 1 year later. The understanding of ENSO response to external forcing is however primitive compared to our understanding of its internal dynamics. There is, for instance, neither a clear consensus on how ENSO is influenced by anthropogenic forcing[5] nor, as we will see below, how it is influenced by volcanism. Some explosive tropical volcanic eruptions are sufficiently strong to inject aerosols into the stratosphere. Those aerosols backscatter incoming solar radiation, and reduce the global surface temperature by a few tenths of a degree Celsius for a couple of years[6]. While this systematic, global effect of large tropical eruptions, such as that of Pinatubo in 1991, is relatively well understood, their impact on regional seasonal phenomena such as the Asian monsoon[7] or on natural modes of climate variability such as ENSO has not yet settled to a consensus[8].

Observational in situ SST data sets dating back to 1882 indicate a large positive ENSO-like pattern following four out of five big eruptions during the historical period (Santa María in October 1902, Mt Agung in March 1963, El Chichón in April 1982 and Pinatubo in June 1991)[9–11]. While this is a small sample, and a warm event was already underway during two of these eruptions, longer ENSO-proxy records also suggest a more probable equatorial Pacific warming within 2 years after large tropical eruptions[12–16].

Modelling studies have however reported contrasting responses to volcanism, ranging from SST anomalies typical of La Niña conditions[10, 17] to no clear signal[18] or preferred El Niño events peaking about a year after the eruption[16, 19–25]. Part of this inconsistency has been attributed to the global volcanically induced surface cooling, which could hide a potential El Niño signal suggested by concomitant positive sea surface height anomalies in the central-eastern Pacific[20]. Modelling studies also do not agree on the mechanisms by which volcanism influences ENSO. Early studies using simple coupled ocean–atmosphere models[26] proposed that following volcano-induced surface cooling, upwelling in the eastern equatorial Pacific acting on a reduced vertical temperature contrast between the ocean surface and interior leads to anomalous warming in this region, thereby favouring El Niño development the following year[12, 27, 62]. In contrast, more complex coupled general circulation models (CGCMs) invoke the combined effect of regional changes in oceanic mixed layer depth, latent heat cooling and cloud feedbacks[10, 19]. Other studies have proposed that the southward shift of the Pacific Intertropical Convergence Zone in response to the Northern Hemisphere cooling[21–24] or the increased land–sea

thermal contrast between the maritime continent and western equatorial Pacific Ocean[19, 25] could drive a weakening of the Pacific trades and lead to an El Niño.

There is hence a need to reconcile in situ observations and paleo-proxies, which suggest an El Niño during the 2 years that follow the eruption, and modelling studies that have reached no consensus on either the sign or the mechanism of the ENSO response to volcanism. Here, we show that using SST anomalies relative to the tropical average (relative SST in the following) allows us to reconcile models and observations. This metric reveals that an El Niño event tends to follow large tropical eruptions in observed SST data sets and in the Coupled Model Intercomparison Project phase 5 (CMIP5) historical experiments. We further use the Institut Pierre Simon Laplace (IPSL)-CM5B[28] climate model and complementary atmospheric and oceanic ensemble simulations, with and without a Pinatubo-like forcing, to identify the main mechanism at play. We choose the Pinatubo eruption in June 1991, as it is the largest and best-documented tropical stratospheric eruption of the instrumental period. We first show that the response to volcanism superposes linearly with the natural ENSO cycle. We then demonstrate that the cooling of tropical Africa (the largest tropical landmass) reduces precipitation, and forces an atmospheric Kelvin wave response that triggers westerly wind anomalies and an El Niño over the Pacific. We also show that, while modulated by the seasonal cycle of convection, the effect of volcanism on wind forcing over the Pacific persists during the second year after the eruption, implying that the Pacific El Niño-like response involves more external forcing than traditional, internally generated events.

## Results

**Relative SST isolates consistent ENSO response to volcanism.** An El Niño event followed four out of five large volcanic eruptions since the beginning of the observational record in the late nineteenth century (Supplementary Fig. 1). This means that the global cooling effect of volcanism is partly masked by the global warming effect of concomitant El Niños over the observing period[29], which led some to conclude that the global post-eruption surface cooling is overestimated in the CMIP5 historical database[8]. In the equatorial Pacific, this aliasing may also explain the physical inconsistency between negative SST anomalies (indicative of La Niña) and concomitant positive sea surface height anomalies in the central-eastern Pacific indicative instead of El Niño[20] after large eruptions. Here we isolate the intrinsic ENSO signal from the volcanically-induced surface cooling by using relative SST (designated RSST) anomalies rather than raw SST anomalies. RSST, which was initially proposed to better explain the response of tropical cyclones to climate fluctuations[30], is defined as the residual signal obtained after removing the mean tropical (20 °N–20 °S) SST anomalies. In observations, RSST (Fig. 1a, b) reveals an equatorial Pacific warming after the five largest tropical volcanic eruptions of the historical period. The resulting RSST composite anomaly in the Niño3.4 region (a common ENSO indicator) reaches 1.1 °C during early boreal winter following the eruptions (Fig. 1a). The probability of having such an average anomaly for 5 randomly picked years is 0.3% according to bootstrap statistics (see Methods and Supplementary Fig. 2). Such low probability is suggestive of a tendency for tropical explosive volcanism to trigger an El Niño event[11].

Using RSST removes the apparent discrepancies among previous modelling studies that have reported responses to volcanism ranging from favouring La Niña to El Niño conditions. For each of the 106 CMIP5[31] historical simulations (see Supplementary Table 1), we identify El Niño events from averaged Niño3.4 region monthly RSST standardized anomalies

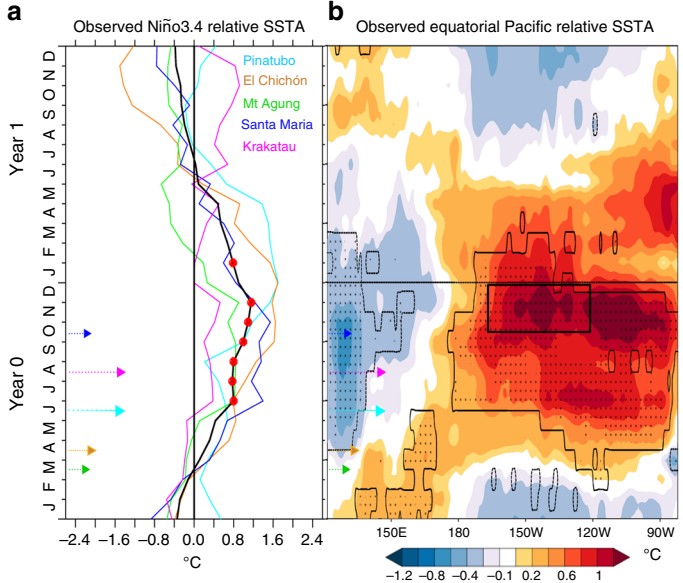

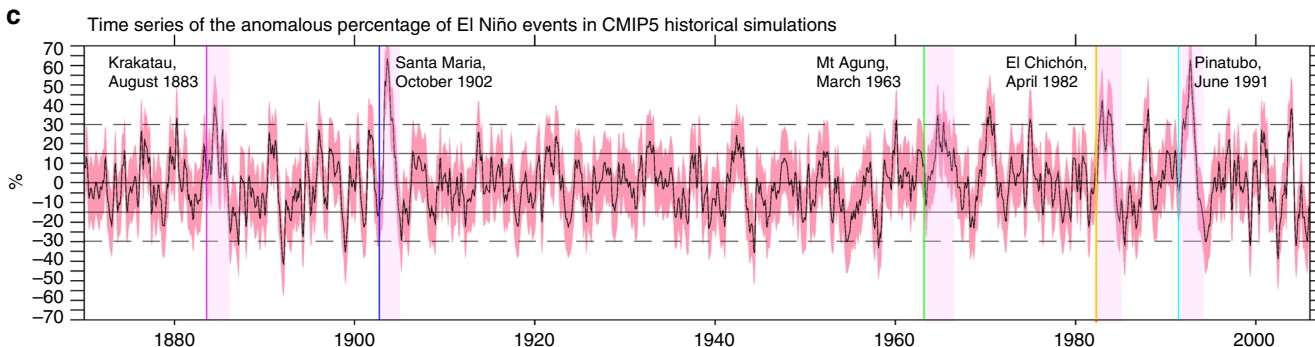

**Fig. 1** Observed and CMIP5 relative sea surface temperature (SST) response to the five main eruptions with stratospheric aerosol injection during 1870–2010. **a** Evolution of the composited Niño3.4 (5 °S–5 °N, 170 °W–120 °W) relative SST anomalies (SSTA) over a 2-year period following the five largest volcanic eruptions during 1870–2010 in HadISST observations[46] (*black line*). The *red dots* denote when the 5 events (*colour lines*) have anomalies of the same sign. An *arrow* indicates the dates and relative magnitude (based on Aerosol Optical Depth provided by Gao et al.[51]) of the selected eruptions. **b** Same as **a** but shown as a longitude-time section. The *black rectangle* denotes the Niño3.4 region during October–November–December. *Stippling* highlights times and locations where the 5 eruptions display anomalies of the same sign and the *contour* corresponds to the 90% significance level of this anomaly according to a two-tailed Student's *t*-test. The composited anomalies are relative to the preceding 5-year climatology. **c** Anomalous percentage of CMIP5 historical simulation members with an El Niño occurrence for the entire historical period (136 years). The anomalous percentage is given relative to the climatological rate over the entire period. El Niño events are defined as Niño3.4 positive relative SST anomalies exceeding 1/2 standard deviation. The *vertical coloured lines* localize each eruption date and the *pink shading* a period of 2 years following the eruption. *Continuous* and *dashed horizontal lines* indicate one and two standard deviations respectively. The *red shading* outlines one standard deviation from the mean of the ensemble

that exceed 0.5 for three consecutive months. This analysis indicates that the number of CMIP5 simulations displaying an El Niño response increases by 30–60% relative to the climatological El Niño probability within 2 years that follows the 5 major volcanic events of the historical period (Fig. 1c and Supplementary Fig. 3a). The low probability ($p < 0.05$; assuming a normal distribution) to have such a high rate of El Niño occurrence during years with no eruption in the CMIP5 historical database indicates a tendency for volcanic forcing to induce El Niños after each major eruption.

The increase of El Niño probability following large tropical eruptions remains undetected when using SST instead of RSST, not only because of the global warming trend, but also because of the volcanically induced global cooling (see Methods and Supplementary Fig. 4). Precipitation anomalies also match RSST much better than SST in the composite response to the Pinatubo eruption built from the CMIP5[31] database (Supplementary Fig. 5), suggesting that RSST better characterizes

the atmospheric convective response to SST gradients and their fluctuations[32, 33]. This underlines the usefulness of RSST for characterizing ENSO response in the context of the global surface cooling induced by volcanic eruptions, which resolves apparent inconsistencies between previous observational[9–16] and modelling[10, 17–25] studies.

**Interaction between volcanism and the natural ENSO cycle.** While there are enhanced chances of El Niño events after the five major historic volcanic eruptions in the CMIP5 database, some members still display neutral or La Niña conditions following these eruptions. This can be explained by the interaction between volcanic forcing and the natural ENSO cycle. As mentioned earlier, the equatorial Pacific Ocean heat content (or WWV) is a well-known ENSO precursor[4]. Using the IPSL-CM5B model we ran three pairs of 3-year-long ensemble experiments, starting from anomalously high, nearly neutral and low equatorial Pacific

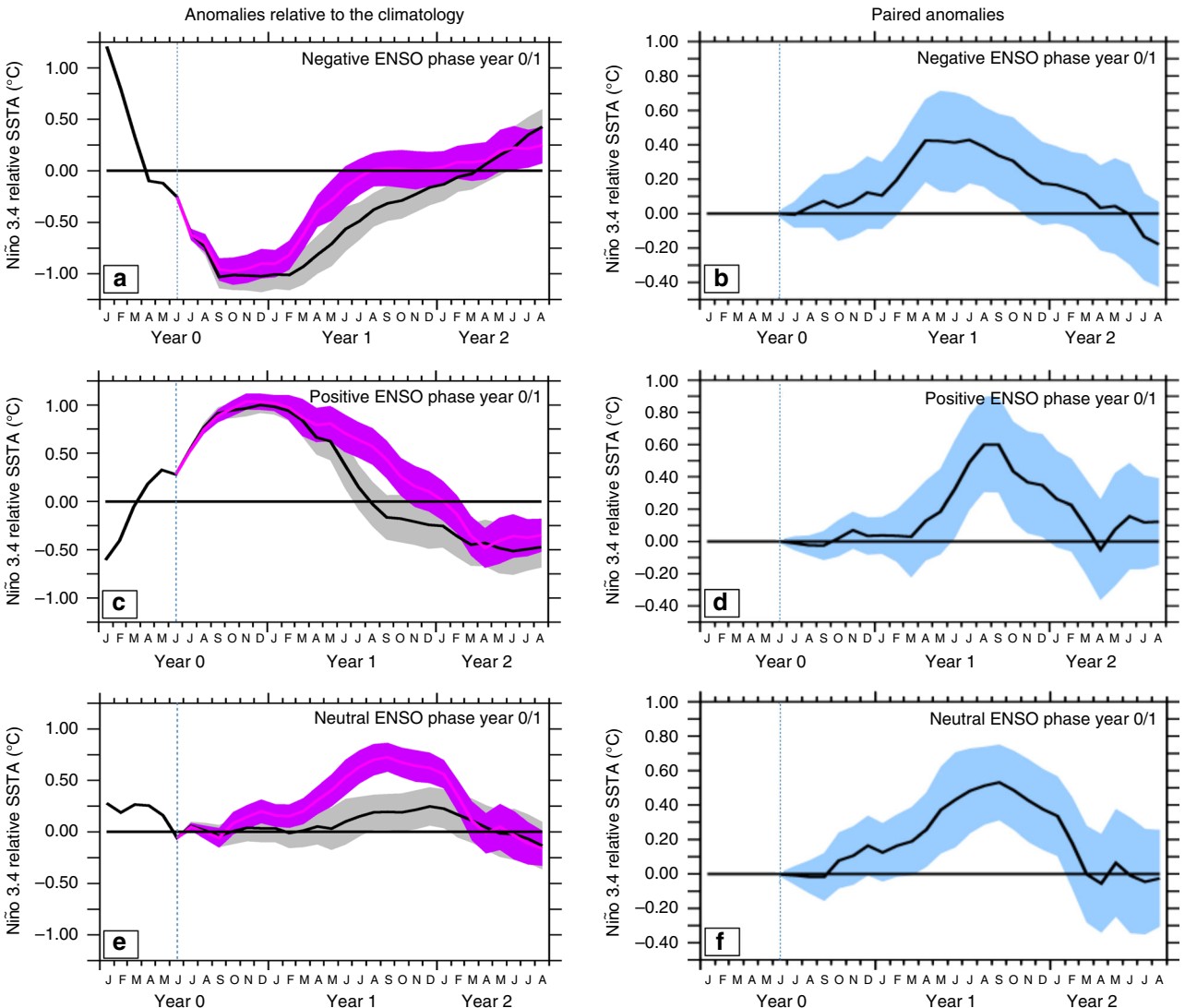

**Fig. 2** Central Pacific relative sea surface temperature (SST) response to the Pinatubo eruption in IPSL-CM5B ensemble simulations. Niño3.4 relative SST (in °C) anomalies, relative to the climatology for the control (*black*) and Pinatubo ensemble mean (*purple*) from January of the year of the eruption (labelled year 0) to August 32 months later, for ocean–atmosphere conditions leading to **a** La Niña, **b** El Niño and **c** neutral ENSO state at the end of the eruption year in the unforced control run, based on the warm water volume (WWV) content. The WWV corresponds to the volume of water above the 20 °C isotherm between 5 °N–5 °N and 120 °E–80 °W and is a good precursor of the upcoming ENSO phase in nature[4]. This is also the case in our model as demonstrated by a peak correlation of $r = 0.7$ when the Niño 3.4 SST index lags the WWV in May by 6 months in the IPSL-CM5B CMIP5 historical simulation[54]. Anomalies are obtained by removing the Niño3.4 relative SST mean seasonal cycle over the 30 years of historical simulations that precede the start of each model ensemble. The *colour shading* indicates the ensemble spread, based on a 90% confidence interval from two-tailed Welch's *t*-test. The *vertical dashed line* in each panel indicates the date of the eruption

heat content shortly before the Pinatubo eruption (see Methods). Each pair consists of a 30-member ensemble with and without the Pinatubo eruption radiative forcing. Paired anomalies between these two ensembles combine the response to volcanic forcing (signal) and internal climate variability (noise), the latter being reduced by averaging across ensemble members.

In the absence of volcanic forcing, anomalously high heat content in late spring favours El Niño, low heat content favours La Niña and neutral heat content favours a nearly neutral state towards the end of the year in our model (Fig. 2), as in observations[4]. Relative to this, the volcanic forcing tends to favour an anomalous central Pacific RSST warming that peaks during the year following the eruption irrespective of the ENSO preconditioning, in line with CMIP5 results (Supplementary

Fig. 3a *light blue curve*) and dedicated studies with other climate models[16, 24, 25] (Supplementary Fig. 6). For ENSO events already underway, volcanic forcing tends to shorten La Niñas (Fig. 2a, b) and lengthen El Niños (Fig. 2c, d). In the absence of preconditioning, volcanic eruptions favour El Niño-like warmings instead of neutral states, but with a warming that peaks in boreal summer and fall rather than boreal winter (Fig. 2e, f).

Figure 3a–f highlights the chain of mechanisms giving rise to an El Niño-like event in the IPSL-CM5B ensemble simulations. Given the relative insensitivity of the ENSO response to the Pacific heat content preconditioning, we have chosen to illustrate these processes for the case of neutral initial conditions (see Supplementary Figs. 7 and 8 for other cases). The tropical Pacific net shortwave reduction at the surface reaches about 10–12 W m$^{-2}$

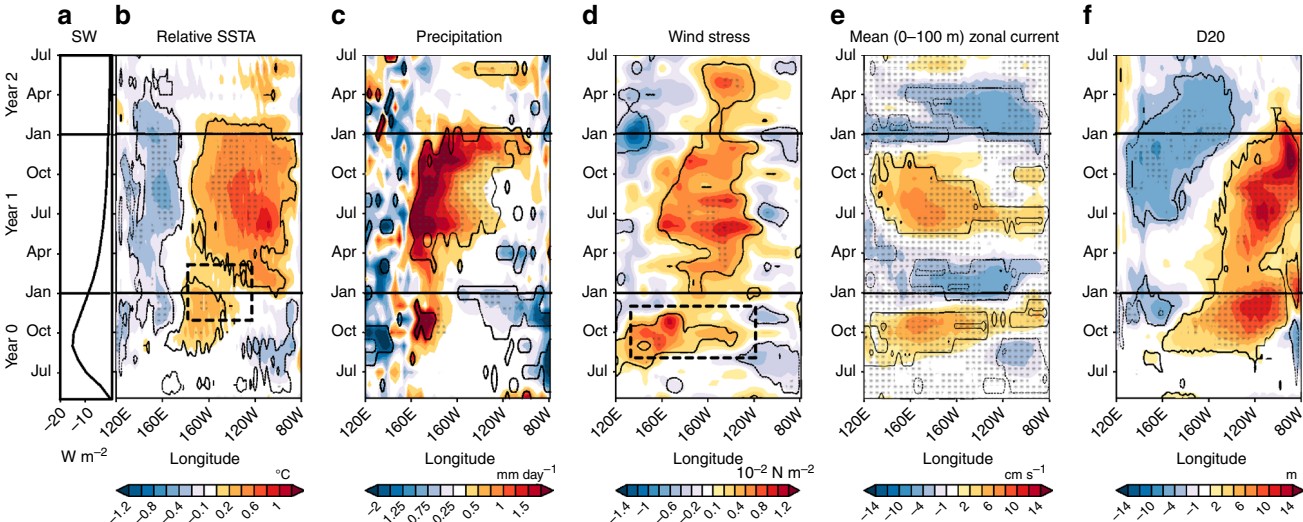

**Fig. 3** Equatorial Pacific response following the Pinatubo eruption. Equatorial Pacific (120 °E–80 °W) clear-sky surface shortwave radiation, relative sea surface temperature (SST), precipitation, zonal wind stress, zonal current and 20 °C isotherm depth (D20) anomalies following the Pinatubo eruption in IPSL-CM5B sensitivity experiments for initial conditions favourable to neutral ENSO state at the end of the eruption year in the unforced control run based on the warm water volume. **a** Time series of mean tropical (20 °S–20 °N) surface clear-sky shortwave anomalies for IPSL-CM5B. Longitude–time section of 5 °S–5 °N anomalous **b** relative SST (°C); the *dashed black box* locates the Niño3.4 (5 °N–5 °S, 170 °W–120 °W) region during November–March; **c** precipitation (mm day$^{-1}$) and **d** zonal wind stress (10$^{-2}$ N m$^{-2}$); the *dashed black box* indicates the region of large-scale westerly wind anomaly in the western Pacific around October of the eruption year, **e** zonal current integrated over the upper 100 m (cm s$^{-1}$) and **f** depth of 20 °C isotherm depth (m) from June of the year of the Pinatubo eruption (year 0) until July 2 years after (year 2). On **e** eastward current corresponds to positive anomalies. *Stippling* on **b–f** indicates time and locations for which at least two thirds of the members display consistent sign anomalies. Contours on **b–f** indicate anomalies that are significantly different from zero at the 90% confidence level, based on a two-tailed Welch's *t*-test

5 months after the eruption (Fig. 3a) as in observations[34], before a gradual recovery until the end of the following year. A large-scale westerly wind anomaly develops in the western Pacific from late August of the eruption year (*dotted frame* in Figs 3d and 4a, b). The resulting downwelling equatorial oceanic Kelvin waves (Fig. 3e) induce anomalous eastward currents and a deepening of the thermocline (Fig. 3f) in the central Pacific, both of which drive a surface warming[35] (Fig. 3b) late in the eruption year (year 0). RSST and thermocline depth anomalies temporarily decrease at the beginning of the following year, consistent with the delayed negative feedbacks usually associated with ENSO demise[1]. Yet, from spring onwards, westerly wind and positive SST anomalies grow again to reach a peak during the year after the eruption.

**Tracking the drivers of the Niño-like response**. The westerly wind anomaly in fall of the eruption year plays an essential role in favouring the development of an El Niño. We have developed targeted ensemble experiments using the atmospheric component of the IPSL-CM5B model forced by the outputs of the coupled model ensemble (see Methods and Supplementary Table 2) to investigate the respective role of the following three processes in its generation. First, we examine the direct effect of volcanic radiative forcing on clouds and atmospheric vapour content[36], potentially altering the atmospheric vertical structure and inducing an atmospheric dynamical response (hereafter referred to as ATM). Volcanic aerosol forcing is included in ATM, but the surface albedo is modified so that continental surfaces do not cool in response to volcanic forcing in this experiment (see Methods, Fig. 4e, f). We also examine the indirect effects of volcanic forcing through the induced horizontal SST gradients[10] (hereafter OCEAN, Fig. 4g, h). The OCEAN experiment includes not only the effect of SST anomalies that develop in response to volcanic forcing[10], but also the positive Bjerknes feedback between the

ocean and atmosphere once El Niño-like SST anomalies have started developing. Continents cool faster than the ocean in response to volcanic forcing, due to their lower heat capacity[6]. The third and last experiment (hereafter referred to as LAND) tests the effect of this differential surface cooling between land and ocean. In LAND, there is no prescribed aerosol forcing as in OCEAN, but the land surface albedo modification enforces a land surface cooling that is consistent with that of the IPSL-CM5B coupled model experiment (Fig. 4i, j). This strategy allows land surface temperature variations in OCEAN experiments, in response to SST anomalies (Fig. 4h), as described in the context of global warming[37, 38]. It is hence more physically relevant than experiments that directly constrain land surface temperature[19], which can result in an exaggerated land cooling and land–ocean gradients. A complementary set of sensitivity experiments (see Methods, Supplementary Table 2) explores the effect of the faster land cooling only over specific regions: tropics (LAND-T), extra-tropics (LAND-ET), Africa (LAND-Africa), Southeast Asia (LAND-SEA) and the maritime continent (LAND-MC).

The equatorial Pacific SST response to surface wind anomalies resulting from each process experiment is then estimated separately by forcing a linear ocean model (see Methods, Supplementary Fig. 9). An additional experiment that includes all three LAND, OCEAN and ATM processes (ALL) reproduces precipitation and surface temperature anomalies simulated by the IPSL-CM5B ensemble simulation (Fig. 4a–d and Supplementary Fig. 10), confirming the relevance of this two-tier approach for understanding how volcanic eruptions induce equatorial Pacific westerlies and RSST anomalies.

**Linking tropical Africa cooling to El Niño onset**. In ATM, the equatorial Pacific wind stress and SST response is negative but hardly significant at the end of the eruption year (Fig. 5a, b *yellow*

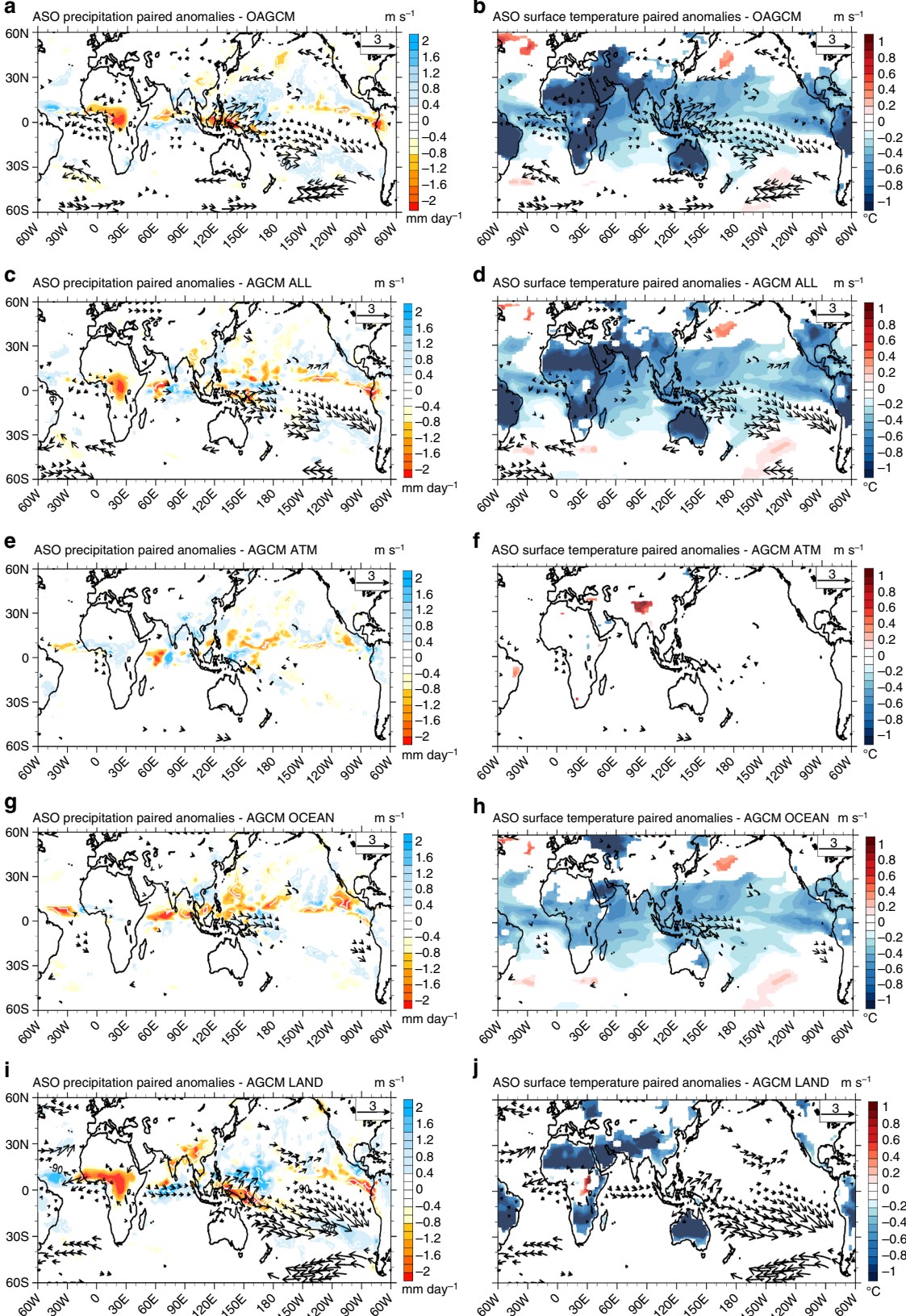

**Fig. 4** Global surface temperature, precipitation and wind patterns in response to the volcanic forcing. August–September–October of year 0 mean precipitation (mm day$^{-1}$, *left*), surface temperature (°C, *right*) and surface wind (m s$^{-1}$, *vectors*) paired anomalies after the Pinatubo eruption in **a**, **b** the IPSL-CM5B coupled model ensemble starting from neutral conditions, and with its atmosphere component initialized with **c**, **d** ALL, **e**, **f** ATM, **g**, **h** OCEAN and **i**, **j** LAND boundary conditions. Anomalies are shown only when significant at 90% according to the two-tailed Welch's *t*-test

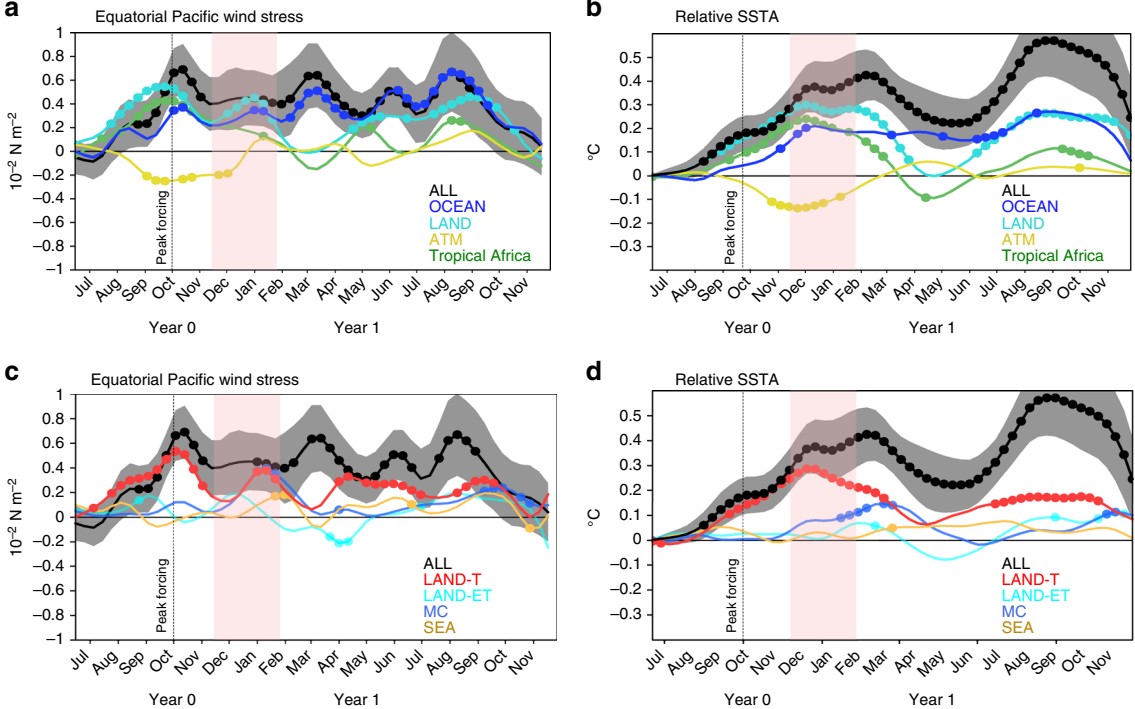

**Fig. 5** Western equatorial Pacific wind stress and relative sea surface temperature (SST) anomalies following the Pinatubo eruption. **a** Time series of averaged western and central Pacific (5 °N–5 °S, 130 °E to 120 °W) wind stress anomalies (in $10^{-2}$ N m$^{-2}$) from ALL, ATM, OCEAN, LAND and tropical Africa atmospheric experiments and **b** the corresponding responses of the Niño3.4 (5 °N–5 °S, 170 °W–120 °W) relative SST (in °C) anomalies in the linear continuously stratified model simulations. **c**, **d** Same as **a**, **b** from the IPSL-CM5B atmospheric component ALL, LAND, LAND-ET, LAND-T, MC and SEA sensitivity experiments. The *dots* indicate months, for which at least two thirds of the individual members display consistent sign anomalies. The *shading* indicates the 90% confidence interval for the ALL experiment only. The *vertical dotted line* and *peak forcing label* in each panel locates the time of the peak surface shortwave anomalies in the tropics (20 °S–20 °N) induced by the volcanic eruption

*curve*), indicating that in isolation from the surface cooling, the direct effect of the radiative forcing on the atmosphere cannot explain the volcanically induced western Pacific initial westerly wind anomaly (Fig. 4e, f) and the subsequent El Niño. The initial westerly wind anomaly after the eruption is instead largely driven by land–sea temperature contrasts, accounted for in the LAND experiment (*light blue curve* in Fig. 5a). These land–sea temperature contrasts induce a strong cooling over tropical continental surfaces (Fig. 4j) in late summer and early fall after the eruption, which results in reduced convective activity and drying of tropical America, Africa and the maritime continent (Fig. 4i). The experiment (see Methods, Supplementary Table 2) including only cooling over tropical Africa indicates that it contributes to more than 80% of the initial westerly wind anomaly (Fig. 5a, *green curve*). Rather than the maritime continent, Southeast Asia or the extra-tropics, it is hence the cooling over the largest tropical landmass, namely Africa, that has the strongest effect on Pacific winds (Fig. 5c). Indeed, the land surface cooling is maximum in the tropics during the first boreal summer and fall (Fig. 4j) and leads to a reduction of the West African monsoon (Fig. 6a, b). The reduced precipitation and tropospheric heating in the equatorial latitudes drive a Matsuno–Gill response[39, 40] where atmospheric equatorial Rossby and Kelvin waves induce easterly wind anomalies over the Atlantic and westerly wind anomalies over the Indian Ocean and western Pacific (Figs. 6a, b and 7 and Supplementary Fig. 11). This Kelvin wave suppresses convection along its path, with reduced convection over the western Pacific that further strengthens the westerly wind signal (Fig. 6a, b). The easterly wind anomalies over the Atlantic induce a shallow thermocline and cold RSST anomalies,

reminiscent of an Atlantic Niña[41], which peak in fall of the eruption year (Fig. 7b). This may also contribute to generating westerly wind anomalies over the Pacific, as noted by recent studies[41, 42].

This western Pacific westerly wind anomaly initiates oceanic downwelling Kelvin waves that warm the central Pacific in winter of the eruption year (Fig. 7b), creating favourable conditions for El Niño development. While the westerly wind and positive RSST anomalies in the Pacific are initially driven by the continental—mostly African—cooling (*light blue* and *green curves* in Fig. 5a, b), the Bjerknes feedback (*deep blue curves* in Fig. 5a, b) strengthens those anomalies during the year after the eruption (Fig. 7c). The Pacific westerly anomalies driven by cooling over the continents weaken during boreal winter of the year following the eruption (Fig. 5a, c), probably because convection moves away from the equator during boreal winter (Fig. 6d–f). During the following year, the cooling of tropical land masses also induces a boreal fall westerly wind anomaly over the tropical Pacific again. This westerly wind anomaly is generated by similar mechanisms to those during the previous year (Fig. 6g–i), although the tropical African cooling and drying play a lesser role, as the tropical volcanic radiative cooling has significantly decreased by this time (Figs. 5a, c and 7c and Supplementary Fig. 12). This volcanically forced westerly wind anomaly also contributes to the second-year Niño-like warming during summer and fall (*dark* and *light blue curves*, Figs. 5a, b and 7d). Overall, this suggests a stronger role of externally forced westerly wind anomalies for volcanically induced El Niño-like events than for internally generated El Niños, for which the Bjerknes positive feedback is responsible for most of the anomalous conditions.

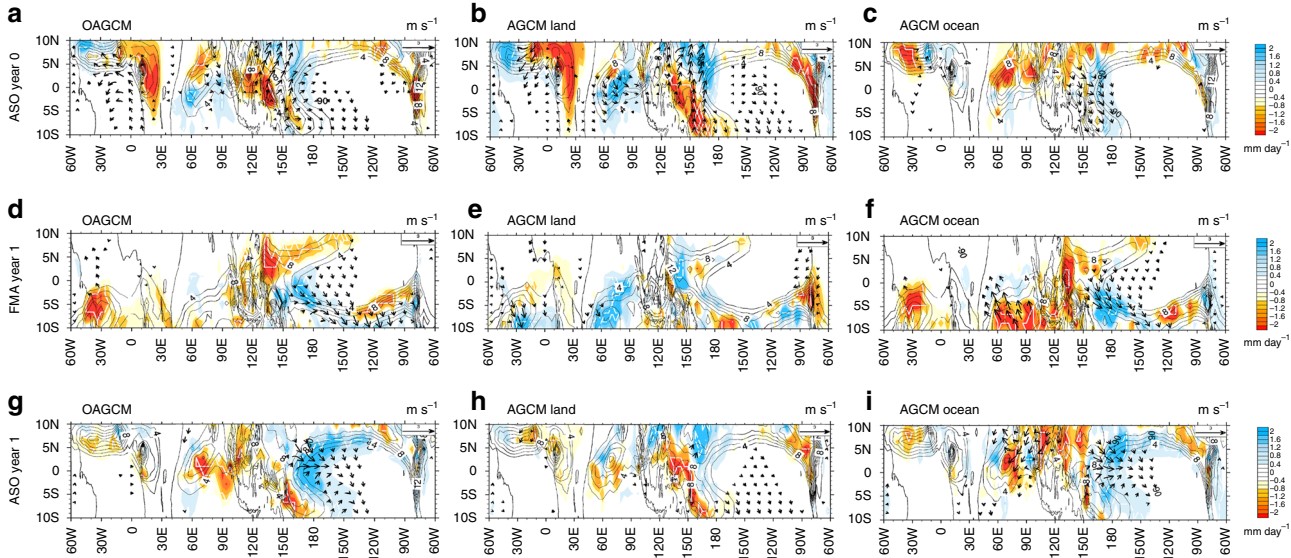

**Fig. 6** Tropical precipitation and wind patterns in response to the volcanic forcing. August–September–October of year 0 (*top row*), February–March–April of year 1 (*middle row*) and August–September–October of year 1 (*bottom row*) tropical (10 °S–10 °N) precipitation (mm day$^{-1}$, *colour shading*) and surface wind (m s$^{-1}$, *vectors*) ensemble mean anomalies resulting from Pinatubo forcing in **a**, **d**, **g** the IPSL-CM5B coupled model ensemble starting from neutral conditions, and the **b**, **e**, **h** LAND and **c**, **f**, **i** OCEAN experiments (see Methods). Anomalies are only shown when significantly different from zero at the 90% confidence level, according to a two-tailed Welch's *t*-test. The *thin black contours* correspond to climatological precipitation (in mm day$^{-1}$) and reveal the seasonal phase locking of Kelvin waves eastward propagation along the latitude of the climatological Intertropical Convergence Zone (ITCZ) and Africa monsoon

## Discussion

Previous work has offered no consensus on how ENSO responds to volcanic forcing because the observational record is relatively short and there were inconsistencies between modelling studies[10–25]. In contrast, our results clearly indicate a tendency for large tropical volcanic eruptions to induce an El Niño-like response peaking during the year after the eruption. Part of the confusion in previous modelling studies arose because they did not properly separate the direct cooling effect of volcanism from that associated with ENSO. Global climate models, for instance, do not overestimate the volcanically induced tropical cooling[43, 44] when accounting for the warm global SST anomalies associated with El Niños[29]. Similarly, it is necessary to remove the volcanically induced tropical cooling signal (for instance through the use of relative SST) in order to clearly isolate the ENSO response to volcanism. This is particularly true for the former studies suggestive of a La Niña response[10, 17], which stumble on the physical inconsistencies between negative SST anomalies (indicative of La Niña) and concomitant positive sea surface height anomalies in the central-eastern Pacific indicative instead of El Niño[20].

Other studies have suggested an alternative mechanism for the influence of volcanically induced land cooling on ENSO. The larger landmass in the Northern Hemisphere for instance may induce a large cooling over Asia and weakening of the Asian monsoon, inducing a southward Intertropical Convergence Zone (ITCZ) shift in boreal summer and fall, and westerly anomalies over the Pacific[21–24]. This mechanism however does not apply in our experiments: the tropical cooling is rather symmetrical with respect to the equator in boreal fall after the eruption (Fig. 4), because the Pinatubo aerosol cloud is still mostly confined within the tropics at that time. Another work has underlined the potential role of the maritime continent cooling in generating the Pacific westerly wind anomalies[19, 25]. Our experiments do not yield any significant summer–fall Asian monsoon weakening or southward ITCZ shift (Fig. 4). Experiments with volcanically induced land cooling only applied over Southeast Asia, the

maritime continent or the extra-tropics do not display any summer–fall westerly wind anomaly in the western Pacific (Fig. 5c). Our modelling results support a different mechanism from those invoked previously, in which the cooling of tropical landmasses, especially Africa, induces boreal summer–fall Pacific westerly wind anomalies that favour a Pacific warming (Fig. 7). This mechanism will however have to be tested in a wider variety of climate models.

Our results may allow further reconciling the CMIP5 models response to volcanism with that from the limited observational record. RSST anomalies peak at the end of the eruption year for most of the observed eruptions, as in our experiments with a recharged equatorial Pacific heat content at the time of the eruption (Fig. 2). Ocean reanalyses[60, 61], only available for the 1991 Pinatubo, 1982 El Chichón and 1963 Agung eruptions, suggest high equatorial Pacific heat content before those eruptions (Supplementary Fig. 13), which would tend to favour maximum RSST anomalies at the end of the year. On the other hand, CMIP5 historical simulations sample the full ENSO cycle (mixture of discharged, neutral and recharged states), and hence yield a response that is more representative of neutral initial conditions.

Our results suggest that volcanically induced El Niño-like events involve a more persistent external wind forcing than classical El Niños, whose growth is dominated by internally generated wind anomalies through the Bjerknes feedback. This may explain the atypical timing of the volcanically induced El Niño-like warming in our experiments that peak in boreal summer rather than boreal winter (Fig. 2). Further analyses are however required to understand how the timing of the eruption influences the El Niño-like response[16] and the link with the West African monsoon and to ascertain the robustness of the above mechanisms in other models. CMIP5 models use very diverse volcanic forcing products and strategies[45] though, and so a unified approach is needed for investigating the effects of volcanism on climate variability, such as proposed in the upcoming CMIP6 VolMIP exercise[8]. An improved understanding and

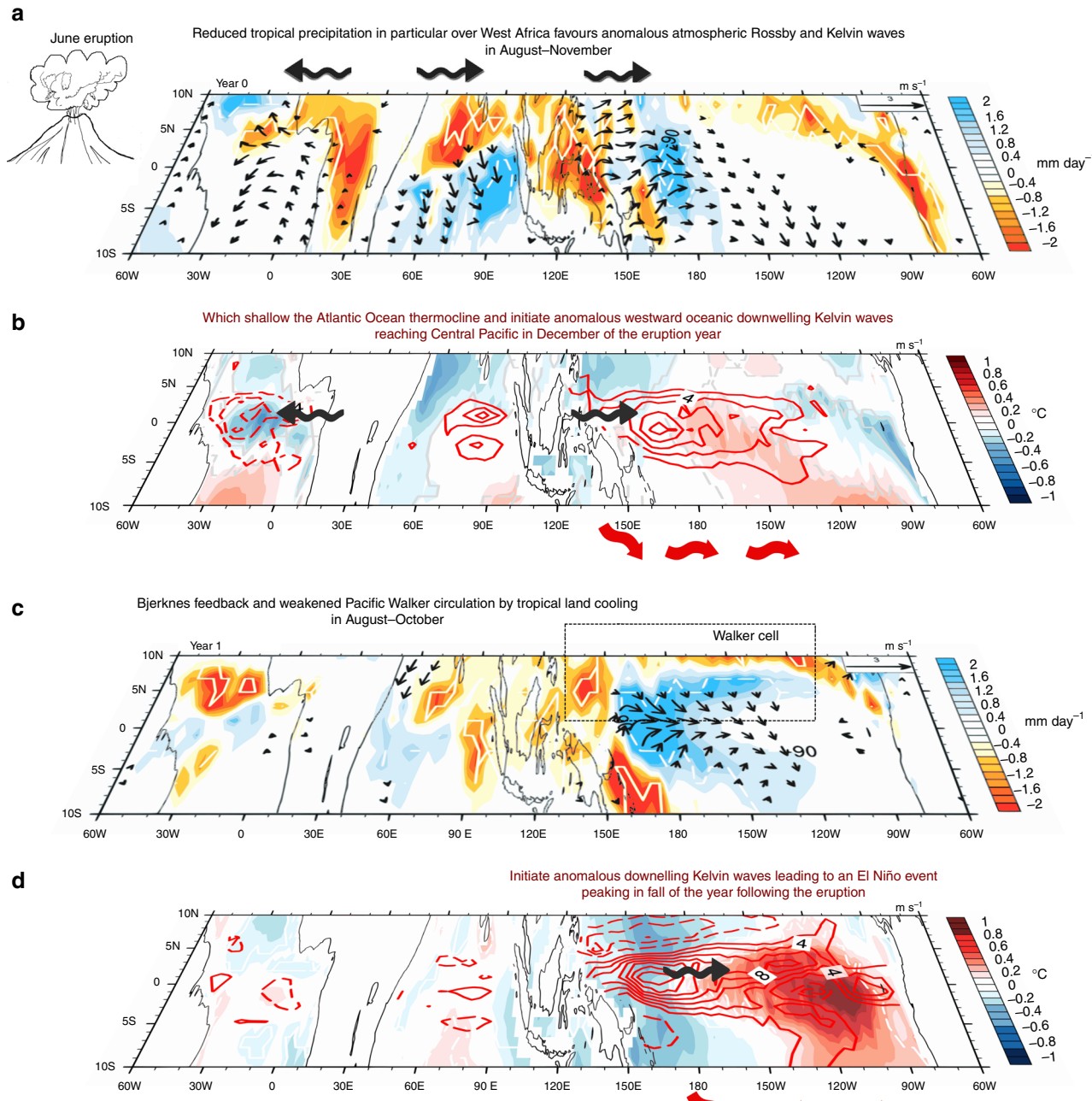

**Fig. 7** Schematic of the mechanism for volcanic El Niño development following a Pinatubo-like eruption in June. **a** September–October–November of year 0 mean tropical precipitation (mm day⁻¹, *shading*) and surface wind (m s⁻¹, *vectors*) anomalies; **b** September–October–November of year 0 relative sea surface temperature (°C, *shading*) and 0–100 m mean zonal ocean current (cm s⁻¹, *red contours*); **c** June–July–August of year 1 mean tropical precipitation (mm day⁻¹, *shading*) and surface wind (m s⁻¹, *vectors*) anomalies; and **d** June–July–August of year 1 relative sea surface temperature (°C, *shading*) and 0–100 m mean zonal ocean current (cm s⁻¹, *red contours*) anomalies after the Pinatubo eruption in the IPSL-CM5B coupled model ensemble for ocean initial conditions leading to neutral ENSO state at the end of the eruption year in the unforced control run based on the warm water volume content. Surface winds are shown when at least two thirds of the individual members display consistent sign anomalies. The *white contours* on precipitation and relative sea surface temperature anomalies indicate anomalies that are significantly different from zero at the 90% significance level according to a two-tailed Welch's *t*-test. Reduced rainfall from August to October of the eruption year induces a tropospheric heating anomaly over equatorial Africa, which forces a Matsuno–Gill characteristic response, with easterly wind anomaly to the west and westerly wind anomaly to east of the heat source, as a result of atmospheric Rossby and Kelvin wave propagation. The downwelling Kelvin wave reduces deep convection along its path, in particular over the western Pacific, which further strengthens the surface westerly wind signal there. The easterly and westerly wind anomalies are represented by the *thick black arrows* in (**a**, **b**, **d**). The *red block arrows* in (**b**, **d**) represent the Pacific Ocean downwelling Kelvin waves response to the westerly wind. The year after the eruption, from August to October, the weaker but persistent tropical land cooling and the Bjerknes feedback both contribute to weaken the Walker circulation (**c**) in the Pacific Ocean, thereby allowing the full development of the El Niño-like event (**d**)

representation of ENSO response to volcanism in climate models may in particular yield extra ENSO predictability after major eruptions, beyond the traditional 6-month lead time limit.

## Methods

**Statistical testing in observations.** We investigated the equatorial Pacific (Niño3.4 and Niño3 regions) SST response to major tropical volcanic eruptions of the instrumental period using composites in HadISST observations[46]. We restricted our analyses to 1870–2010 as there are too many gaps in spatial sampling before 1870. The earliest part of the record (late nineteenth century) is, however, less reliable due to lower spatial resolution. We build composites using the five tropical volcanic eruptions in 1870–2010 with significant injection of sulphuric aerosol in the tropical stratosphere: Pinatubo (June 1991), El Chichón (April 1982), Mt Agung (March 1963), Santa María (October 1902) and Krakatau (August 1883). SST anomalies are computed relative to the 5-year mean preceding each eruption (to remove multidecadal variability and long-term trends) and smoothed with a 3-month Hanning filter (to filter out intraseasonal variability). To evaluate the level of statistical significance, we first count the number of events with anomalies that have the same sign, as shown in Fig. 1a. If we assume that there is the same probability of having positive and negative signs, according to the binomial law, the likelihood for having 5 anomalies of the same sign is 0.03, i.e., corresponding to a 97% confidence level. Second, we use a Monte Carlo approach to compute the distribution for the composite value of November–January RSST average anomalies of 5 randomly picked years within the historical period, the result of which is given in Supplementary Fig. 2. The probability in the Niño3.4 composite of RSST anomaly equal or above the observed 1.1 °C value is very small (0.3%). The level of statistical significance is similar in other data sets such as HadiSSTv1.1, HadSST31, ERSSTv2b2[47], Kaplan SST3[48] and Niño34hist4[49]. Various methods to compute the anomalies (15-year high-pass filter, removing the mean value for the previous 1, 3, 5 or 10 years) were tested and did not affect the results.

**CMIP5 historical simulations.** We considered 26 CMIP5 models and a total of 106 members from the CMIP5 historical simulation database[31] (Supplementary Table 1). Each member is a coupled ocean–atmosphere simulation using the CMIP5 recommended transient forcing from 1850–2005. We did not include simulations using anomalies of the solar constant to mimic volcanic forcing, interactive chemistry, carbon cycle or prognostic global aerosol models in our analyses, to restrict the potential noise associated with those extra degrees of freedom. Despite this, there is still a relatively large diversity of specified external forcings, as shown in Supplementary Table 1. Anomalies were computed relative to the 1961–1990 climatology of each model simulation. We analysed SST evolution in these simulations by using both the raw and relative (obtained by removing the global tropical 20 °N–20 °S mean SST) SST anomalies. For each of the 106 CMIP5 historical simulations[31], we identified El Niño events from averaged Niño3.4 region monthly raw and relative SST standardized anomalies that exceed half a standard deviation over the 1870–2010 simulated period. The percentage of CMIP5 simulations displaying an El Niño-like state and its statistical significance are shown in Supplementary Figs. 3–5. Using various methods to compute the SST anomalies (removing the mean climatology of the previous 5 years, using the 1961–1990 or the 1870–2010 baseline climatology) do not affect the results. The difference between raw SST (Supplementary Fig. 1a, b) and RSST (Fig. 1a, b) is weak in observations, but it is large in CMIP5 models (see Supplementary Fig. 5). The largest uncertainties in the estimates of radiative forcing from CMIP5 historical simulations occur during periods of volcanic activity[43] and generally models tend to overestimate the observed post-eruption global surface cooling[44]. Relying on relative SST and precipitation anomalies partly corrects this bias, so that it reveals the dynamical ENSO response to volcanic forcing in CMIP5 models.

**Coupled model simulations for the 1991 Pinatubo eruption.** To help analyse the response of ENSO to volcanic forcing, a suite of short ensemble experiments was run over January 1991 to December 1993 for the Mt Pinatubo eruption (June 1991) using the IPSL-CM5B-LR[28] ocean atmosphere CGCM. The stratospheric aerosol optical depth (hereafter AOD) forcing due to the Pinatubo eruption in each member uses the Gao et al.[51] data set. The stratospheric volcanic cloud spreads from the tropics to the poles from the start of the eruption in June 1991 until late 1992, and then slowly disappears as the aerosols are totally washed out of the atmosphere in 1993. The total AOD reaches its maximum in boreal autumn–winter 1991–1992. Corresponding control ensembles starting from the same initial conditions but without Pinatubo volcanic forcing were performed.

**IPSL-CM5B-LR coupled model simulations ensemble.** We use the IPSL-CM5B-LR CGCM. The Laboratoire de Météorologie Dynamique (LMD) developed the LMDZ5B atmospheric component, which is described in Hourdin et al.[52]. This model uses hybrid $\sigma$–$p$ vertical coordinates, with 39 pressure levels including 18 in the stratosphere. The LMDZ5B model uses ORCHIDEE as land surface component. We used the low-resolution grid with 1.87° in latitude and 3.75° longitude. The oceanic component, Nucleus for European Modeling of the Ocean (NEMO) version 3.2, is an oceanic GCM with 31 vertical levels (whose thickness varies from

10 m near the surface to 500 m towards the bottom), a 2-level representation of sea ice and a mean spatial horizontal resolution of about 2° (with a refinement of the latitudinal resolution to 0.5° near the equator; ORCA2 grid). Previous works[53] found that this model reproduces ENSO variability with two spectral peaks around 3–3.5 years and beyond 4 years, which is in good qualitative agreement with observations.

The observed ENSO seasonal phase locking with a peak occurring mostly in boreal winter (November to January) and feedbacks are also well captured in this model[53]. Initial conditions were chosen in the IPSL-CM5B-LR standard historical run. To disentangle the role of Pinatubo eruption on ENSO evolution and avoid model spin-up, we chose restart days in the historical run with greenhouses gases and tropospheric aerosols concentrations close to the 1990s levels and more than 5 years after any volcanic eruption. We selected three restart dates corresponding to 1 June (i.e., shortly before the Pinatubo eruption) with initial conditions that would be favourable to the development of an El Niño, neutral, or La Niña phase in the absence of volcanic forcing. WWV (the volume of water above the 20 °C isotherm within 5 °N–5 °N 120 °E–80 °W) is a good precursor of the upcoming ENSO phase in nature[4]. This is also the case in our model as demonstrated by a peak correlation of $r = 0.7$ when the Niño 3.4 SST index lags the WWV in May by 6 months in the IPSL-CM5B CMIP5 historical simulation[54]. We hence selected 3 model states representing discharged, near-neutral and recharged WWV based on normalized WWV values in May. For the three initial states, 30 members (forced and unforced) were generated by adding a small white noise on the initial SST. Consistent with observations, the discharged, neutral and recharged states ensemble mean respectively produce a La Niña, near-neutral and El Niño state in the absence of volcanic forcing (see Fig. 2 and Supplementary Fig. 7).

**CNRM-CM5 coupled model simulations ensemble.** To evaluate the robustness of the results obtained with the IPSL climate model, we also used the CNRM-CM5 model developed jointly by CNRM-GAME (Centre National de Recherches Météorologiques—Groupe d'études de l'Atmosphère Météorologique) and Cerfacs (Centre Européen de Recherche et de Formation Avancée). CNRM-CM5, described by Voldoire et al.[50], couples the ARPEGE-Climat (v5.2) atmosphere, NEMO (v3.2) ocean, the ISBA land surface and GELATO (v5) sea ice models through the OASIS (v3) coupler. The horizontal resolution is 1.4° in the atmosphere and 1° in the ocean (with a refinement of the latitudinal resolution to 0.5° near the equator). The model reproduces ENSO typical variability well with an ~2-7-year period and a seasonal phase locking with a peak occurring mostly in boreal winter, in good qualitative agreement with observations[53]. The ensemble of the CNRM-CM5 model was designed to explore the role of Pinatubo eruption on ENSO response when starting from random initial conditions. The initial dates were chosen randomly in the control and historical runs performed with the same model version[50].

**The two-tier sensitivity experiments approach.** To explore the role of various processes that drive the climate response to volcanism, we ran two-tier experiments using an atmospheric model simulations ensemble (AGCM) and a linear ocean model.

**The atmospheric model simulations ensemble.** We ran several 30-member AGCM ensembles using the atmospheric component of the IPSL-CM5B coupled model[52]. Daily fields from each of the 30 members of the coupled model ensembles starting from neutral initial conditions are used to derive a set of boundary conditions (described below) for the stand-alone atmospheric component (LMDZ AGCM) of the IPSL-CM5B model, which includes the ORCHIDEE land surface module. The stand-alone atmospheric model is run for 2 years from 1 January 1991 onward (i.e., year 0; 5.5 months before the Pinatubo eruption) to enable the atmosphere to dynamically adjust to the boundary conditions. We have first run a Control 30-member AGCM ensemble using surface boundary conditions (i.e., SST and sea ice cover) from the coupled model control ensemble without including the changes induced the volcanic aerosol. The ALL experiment follows the same protocol, but includes the stratospheric AOD evolution corresponding to the Pinatubo eruption as in the coupled model experiments and specified SST and sea ice cover from each member of the coupled model Pinatubo ensemble. The anomalies of ALL relative to the Control are almost identical when compared to the anomalies of the coupled model experiment (Supplementary Fig. 10). The small differences between the coupled and the atmospheric experiment are explained by the differences in the surface flux due to coupling[55]. To address the role of the atmospheric, land and ocean responses in triggering surface wind anomalies, five complementary sets of experiments have been run (see summary in Supplementary Table 2).

The ATM, LAND and OCEAN experiments were respectively designed to represent the effect of aerosol forcing through its direct impact on the atmospheric structure (ATM), and indirect impacts through changes in land (LAND) or ocean (OCEAN) surface temperatures. The ATM and LAND scenarios were developed using the same protocol as in ALL but with prescribed daily SST from the Control members (i.e., with a SST that excludes the impact of volcanic aerosol forcing). Volcanic aerosol forcing is included in the ATM experiments, but the surface albedo in this experiment is modified so that continental surfaces do not cool in response to volcanic forcing (details below). In LAND, there is no prescribed

aerosol forcing, but the albedo modification enforces a land surface cooling that is consistent with that of the Pinatubo coupled model experiment. This strategy has the advantage of allowing surface land temperature to vary in the presence of SST anomalies (in the OCEAN experiment)[37, 38], as is seen in other model experiments in the context of global warming scenario[37, 38]. This is hence more physically relevant than experiments that constrain land surface temperature[19] and which can result in unphysical and exaggerated land cooling, and hence exaggerated land–ocean gradients. We modify the land surface albedo to restore the atmospheric temperatures over land to that of the Control coupled model ensemble in ATM and to that of the Pinatubo coupled model ensemble in LAND.

The boundary layer temperature changes $\Delta T$ in response to a given change in the net shortwave radiation $\Delta SWnet_{SFC}$ at the surface can be approximated assuming linear dependency such as:

$$\lambda \Delta T = \Delta(SWdown_{SFC} - SWup_{SFC}) = \Delta SWnet_{SFC}$$

with $\lambda$ being the feedback parameter including that due to the Planck emission (i.e., the feedback associated with changes in surface temperature), changes in vertical temperature lapse rate, water vapour and clouds. Here, we specify $SWnet_{SFC}$ by prescribing the surface albedo $\alpha$. The surface albedo is related to the net shortwave radiation by:

$$SWnet_{SFC} = SWdown_{SFC}(1 - \alpha) \tag{1}$$

In particular, we have:

$$SWnet_{cSFC} = SWdown_{cSFC}(1 - \alpha_c) \tag{2}$$

$$SWnet_{pSFC} = SWdown_{pSFC}(1 - \alpha_p) \tag{3}$$

Here, the subscript c denotes the values of the control coupled model experiment, while the subscript p designates the values of the Pinatubo coupled model experiments. Then, we calculate $\Delta\alpha$, the relative change of the incoming SW radiation due to stratospheric sulphate aerosols:

$$\Delta\alpha = (SWdown_{pSFC} - SWdown_{cSFC})/SWdown_{cSFC} \tag{4}$$

In ATM, the downward shortwave radiation is modified due to the direct radiative effect of stratospheric sulphate aerosols (i.e., $SWdown_{SFCATM}= SWdown_{pSFC}$), but we modify the surface albedo to $\alpha_{ATM}$ so that the surface-absorbed net shortwave radiation remains equal to that of the control run (i.e., $SWnet_{SFCATM}=SWnet_{cSFC}$). Using previous relationships together with (2) and (4), we obtain:

$$\alpha_{ATM} = (\alpha_c + \Delta\alpha)/(1 + \Delta\alpha) \tag{5}$$

Conversely, in LAND, the surface albedo is set to the $\alpha_{LAND}$ value, with downward shortwave radiation equal to the control experiments ($SWdown_{SFCLAND}= SWdown_{cSFC}$), but surface-absorbed shortwave equal to that of the Pinatubo experiments (i.e., $SWnet_{SFCLAND}=SWnet_{pSFC}$):

$$\alpha_{LAND} = \alpha_p - \Delta\alpha + \alpha_p\Delta\alpha \tag{6}$$

We diagnosed $\Delta\alpha$ from the control and Pinatubo mean ensemble simulations and performed ATM and LAND simulations imposing the albedo in order to restore the surface temperature towards control and Pinatubo values respectively. This protocol involves a linearity assumption of assumed constant values for the feedback parameter $\lambda$. Surface temperature anomalies in the ATM, LAND and OCEAN experiments (Fig. 4 and Supplementary Fig. 10) however add up quite linearly to surface temperature anomalies in the ALL atmospheric or Pinatubo coupled model experiment, indicating the validity of our approach. The LAND experiment isolates the effect of volcanically induced continental cooling but does not narrow down the region that plays a key role for inducing wind anomalies over the tropical Pacific. We hence also developed a suite of additional 30-member LAND sensitivity experiments (with and without Pinatubo radiative forcing) that allows exploring the respective roles of land cooling in the tropics (LAND-T), the extra-tropics (LAND-ET), Africa (LAND-Africa), Southeast Asia (LAND-SEA) and the maritime continent (LAND-MC) only (See Supplementary Table 2).

**The linear Indo-Pacific Ocean model**. The atmospheric experiments above isolate the wind changes over the Pacific associated with three different processes: direct effect of radiative forcing on the atmosphere (ATM) and indirect effect through the cooling of continents (LAND) and induced oceanic SST gradients (OCEAN). To translate those wind signals into SST responses, we use a linear continuously stratified[56] (LCS) ocean model of the Indo-Pacific region combined with a linear SST equation. The LCS model is forced by the atmospheric forcing produced by each of the 30-member experiments above to deduce the equatorial Pacific Ocean response (and in particular the SST response). The LCS model for the Indo-Pacific Ocean resolves vertical baroclinic modes. It is used to simulate thermocline depth and near-surface current anomalies. It is combined with a linear SST equation[57] to realistically resolve the variations of SST due to wind stress anomalies:

$$\frac{dSST'}{dt} = -U'\frac{d\overline{SST}}{dx} - V'\frac{d\overline{SST}}{dy} + \gamma H' - \beta SST'$$

The horizontal advection term is approximated from the anomalous advection of the climatological SST ($\overline{SST}$) gradient (here from the IPSL historical run climatology) by the LCS 0–100 m averaged anomalous currents $U'$ and $V'$. In the Indo-Pacific region, zonal advection dominates over meridional advection (not shown, see for example Vialard et al.[35]). The $\gamma H'$ term ($H'$ is the thermocline depth anomaly obtained as $H'=150$ $SSH'$ from the sea surface height anomaly $SSH'$) parameterizes the vertical physics, i.e., the interannual modulation of mixing and upwelling by thermocline depth anomalies. The coefficient $\gamma$ is constant to the east of 140 °W ($\gamma=0.028$ °C/month, as in Burgers[58]), and linearly decreases until 1/5 of its maximum value at the western boundary of the Pacific, as in McGregor et al.[59]. This mimics the decreasing effect of the upwelling as the climatological thermocline deepens towards the west. Heat fluxes are parameterized as a Newtonian damping $-\beta SST'$, with the associated damping timescale $\beta^{-1}=2$ months, close to values used by McGregor et al.[59] and Burgers[58]. The 5 first baroclinic modes are used (even though mainly only the first two modes contribute)[57]. This model has the advantage of being relatively realistic (Validation: high correlations of 0.8–0.9 with observations (TAO, AVISO, OSCAR), notably for zonal current and SST in Niño4 (central Pacific), for which the first two baroclinic modes are important, and also for SST and SSH in Niño3 (eastern Pacific) when the model is forced by ERAI (see details in Izumo et al.[57], as well as rather simple and cheap to use. It is used to simulate the SST response to wind stress anomalies diagnosed from the ATM, LAND and OCEAN AGCM ensemble members. The high correlation of the IPSL-CM5B coupled model Niño3.4 SST anomalies with those simulated by the LCS model forced by the coupled model experiments ($r = 0.9$; Supplementary Fig. 9) and ALL ($r = 0.8$, not shown) ensemble wind stress justifies our approach. Furthermore, the favourable comparison of the ALL ensemble mean with the coupled model experiment wind stress anomalies in the western equatorial Pacific region (Supplementary Fig. 10) lends credence to our method and suggests that our two-tier methodology to investigate various physical mechanisms in forced experiments derived from the coupled one is sound.

**Estimating signal-to-noise ratios**. For the analysis of these Pinatubo sensitivity experiments, we used two approaches to evaluate the statistical significance of anomalies associated with volcanic forcing, relative to purely random, internal coupled ocean–atmosphere variability. The anomalies associated with the response to volcanism are computed as paired differences between each of the members of the Pinatubo experiment and its unforced counterpart. The first statistical calculation estimates the statistical significance of the anomaly between the two ensemble means using a two-tailed Welch's $t$-test, and taking into account the size and variance of each (control and Pinatubo) ensemble. For CMIP5 model analyses, the significance of anomalies relative to the climatology is measured using a two-tailed Student's $t$-test, considering each model as an independent degree of freedom. For both our paired Pinatubo ensembles and CMIP5 models, we also estimated at each time step and grid point the percentage for which at least two thirds of the individual simulations have anomalies of consistent sign, which is a commonly used technique for analysing outputs of CMIP models. This allows us to verify that statistically significant anomalies in the ensemble mean are representative of the ensemble members and not due to a few extreme members. Figure captions specify the test for each figure.

**Code and data availability**. The code and data that support the findings of this study are available from the corresponding authors on request.

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

## Acknowledgements

We thank three anonymous reviewers for their constructive criticisms of an earlier version of this manuscript. We also thank Felix Bourtourault for the volcano drawing on Fig. 7. This research was supported by a grant from the LABEX L-IPSL, funded by the French Agence Nationale de la Recherche under the Programme d'Investissements d'Avenir (Grant no. ANR-10-LABX-18-01), by a grant from the Agence Nationale de la Recherche MORDICUS under the Programme Environnement et Société (Grant no. ANR-13-SENV-0002-02) and by a grant from European Commissions Seventh Frame-work Research Program (FP7) under the grant agreement 308378 (SPECS). It also benefited from the IPSL CMIP data access PRODIGUER. A.R. is supported by the US National Science Foundation grant AGS-1430051 and M.J.M. is supported by NOAA. This is PMEL contribution 4492.

## Author contributions

M.K. and J.V. designed the study with inputs from T.I., C.C., S.J., G.G., E.G., J.M., M.L., N.L., A.R. and M.J.M. M.K. developed the experimental strategy and ran the GCM

models. T.I. performed the linear, continuously stratified, model simulations and the probability density function of observed SST anomalies. M.K. performed the observed and CMIP5 superimposed epoch composites over the historical period and analysed the model simulation outputs. All authors contributed to the interpretation and writing.

## Additional information

**Competing interests:** The authors declare no competing financial interests.

**Change history:** A correction to this article has been published and is linked from the HTML version of this paper.

