## [Peer Review file · Nature Communications]

Reviewers' comments:

Reviewer #1 (Remarks to the Author):

The authors investigate the ENSO's response to tropical volcanic eruptions both (i) using CMIP5 models, and (ii) performing a set of sensitivity simulations using two different climate models (IPSL-CN5A-LR and CNRM-CM5). The authors show that the relative SST (RSST) - instead of SST - should be used to detect the El Niño-like response after volcanic eruptions in CMIP5 model. The authors then point to westerly wind anomalies in the fall following the eruption, as triggers for El Niño-like response. They claim that the wind anomalies are caused by a giant sea breeze generated by the stronger cooling of the Maritime Continent compared to the equatorial Pacific following a tropical eruption.

The results are potentially of interest for the Nat. Comm. broad audience. However, I have several concerns that prevent me to recommend it for publication in its present form.

Major comments:

1) THE MODEL ADOPTED:

a. The authors used two climate models to test their hypothesis. The models used are IPSL-CN5A-LR and CNRM-CM5. The authors highlight in the supplementary material that the IPSL model has some issues with the ENSO seasonality. Hence, IPSL-CN5A-LR is a concern and should not be used for studies investigating changes in ENSO. Not only IPSL-CN5A-LR has an ENSO seasonality peaking in the wrong season, but also all the feedbacks associated with the ENSO dynamics are largely off (see Fig. 3 in Bellenger et al., 2014 - IPSL (a)). It is surprising that the authors did not adopt the IPSL-CN5B-LR model instead (see IPSL (c) in Fig. 3 in Bellenger et al., 2014), which seems to perform better. On the other hand, the CNRM model seems to have a good representation of ENSO mean climate and feedbacks.

Therefore, I would not recommend to use IPSL-CN5A-LR, but instead, to switch to IPSL-CN5B-LR. For instance, figure 2 clearly shows a notable different evolution in the SST anomalies following the volcanic eruptions, which may simply be a result of the wrong ENSO representation in the IPSL model.

b. The authors opted to use IPSL to CNRM to perform the uncoupled sensitivity simulations; however, no justification is presented regarding this choice. I am puzzled about this choice, since in the sensitivity experiments the authors need to prescribe the SST, which has a wrong seasonality in the Tropical Pacific.

2) THE APPROACH:

a. It is not clear to me how the authors created the Pinatubo and unforced ensemble members. The authors state that the unforced members "were performed, only differing from the forced ensemble by the specified Pinatubo volcanic forcing." In that case, the first 5 months (Jan-May) should be identical in the two ensembles (forced and unforced) since the eruption occurs in June. However, Figure 3 clearly shows that they do not have the same initial conditions. Even if a smoothing (3 or 5 month running mean) is applied to the Niño3 index at least the first 3-4 months should be identical or barely distinguishable, which is clearly not the case.

In case, the authors - for unknown reasons - have used slightly different initial conditions in January, in order to initialize the unforced ensemble members relative to the volcano members, the simulations should be re-run: the benefit of running your own simulations compared to CMIP is because you can clearly disentangle the role of volcanic eruption performing volcano and unforced run starting from identical initial conditions.

b. The authors blend together the results from the two models in Figure 3, which is quite misleading. This is particularly important, as in one of the models the ENSO seasonality is completely off, peaking in spring/summer. The authors need to keep the models' results separated

if they will still use IPSL-CN5A-LR.

3) THE MECHANISM:

The authors show that El Nino response starts with a wind stress anomaly in October. They perform a set of sensitivity studies to try to understand the cause of such wind stress anomaly in the western tropical Pacific. The authors conclude that the LAND experiment (i.e. when they just change the surface temperature over land) is able to reproduce the westerly wind anomaly in the model. The authors explain the mechanism based on relatively local (regional) scale changes, without investigating large-scale changes that could have triggered it. I am skeptical that the cooling (which is barely significant and of the order of $0.2\text{ }^{\circ}\text{C}$) of the Maritime Continent alone could trigger the chain of events described by the authors. They show the change in temperature over the Maritime Continent (no specification of exact lon and lat used) and Western Equatorial Pacific SST in Figure S5; however, the western boundary of the Western Equatorial Pacific domain considered by the authors is 90°E , i.e. west of the entire Maritime continent). Since the authors are claiming that the difference in temperature is causing the giant breeze, the two areas (land vs. ocean) should therefore not overlap and they should consider a further eastern domain (e.g. $120 - 180^{\circ}\text{E}$).

On the other hand, a recent study (Pausata et al., 2015) showed that the NH cooling would lead to a shift south of the ITCZ in the Pacific, which would weaken the trades. They also showed that for an eruption occurring in summer, the surface wind changes and the ITCZ shift over the Pacific would occur around September/October, because of lower response over the Pacific Ocean relative to the Asian continent were the shift already started in the summer with a weakening of the Asian Monsoon system (AMS).

They simulated high-latitude volcanic eruptions; nevertheless as the authors themselves pointed out the land will cool more rapidly than the ocean, i.e. the NH is likely to cool more than the SH. Hence, the same mechanism can be at play in your simulations. The weakening of the AMS and/or the ITCZ shift over the Pacific could be the cause of the westerly anomalies also in your case. Recently, Stevenson et al. (2016) also mentioned the ITCZ shift as trigger of the El Nino-like response.

To rule out which kind of mechanism is driving the El Nino-like response, an additional sensitivity test is required, in which the cooling is applied only to Southeast Asia.

The authors should show and quantify the cooling as well as potential shift in the ITCZ and discuss their results in the context of previous studies (Pausata et al., 2015 and Stevenson et al., 2016). Hence a map showing the global cooling and precipitation changes in different seasons (JJA, SON, DJF) would be beneficial.

Finally, although I'm not familiar with the method used by the authors for the uncoupled sensitivity test (albedo change), the authors should show, for instance, that the cooling simulated over land is comparable, both in the LAND experiment and in the fully coupled experiment.

Minor comments:

- The authors mentioned that the proxy archives suggest the occurrence of El Nino-like conditions following volcanic eruptions. However, to my knowledge, this situation has not yet been fully clarified using proxies. For instance, D'Arrigo et al., 2009 and Anchukaitis et al., 2010 showed a La Nina like response after volcanic eruptions. Therefore, based on previously published studies, I would be cautious stating that proxy records indicate El Nino conditions after volcanic eruptions.

- Given the broad NCOMMS audience I would suggest to use the term Southeast Asia rather than Maritime Continent that is less known (or specify the geographical region).

- LL 47: the authors state that the RSST anomaly in Niño3.4 region reaches 1.1°C , that has a 0.3% probability as shown in Figure S1. However, if I understood correctly Figure S1,

this figure shows the pdf of averaged NDJ RSST anomalies, while the 1.1{degree sign}C anomaly refers to just one month. The 3-month average should probably be around 0.9/0.95{degree sign}C; hence, the authors should correct the probability (~0.7-1%).

By the way, it is quite surprising that the largest eruption considered in this study does not produce a warm anomaly. Related to that I question the choice of using 1961-1990 as climatology! What is the simulated trend in SST from 1880 to 2000? I would expect 0.3/0.4{degree sign}C warmer SST in the tropical Pacific (NINO region) in the 1961-1990 compared to the decade of the Krakatau eruption 1880-1889.

- The authors show that RSST is better than SST in capturing the El Nino response in CMIP5 models. However, I would be very curious to see the same plot in figure 1a and b done using SST instead of RSST. This figure should go in the supplementary to support that statement.

- Fig. 1: In panel a, I assume 0.6{degree sign}C should actually be 0.8{degree sign}C. Please correct it or use a linear scale.

- L62: 0.75 what? Standard deviation? Is this the common metric used when analyzing CMIP5 models? In observation the 0.4{degree sign}C threshold (exceeded for at least 6 consecutive months) is used for Nino3.4 which is 0.5 standard deviation.

- LL64-67 and Fig. 1c: I must stay I haven't followed exactly how that analysis has been performed. Could the authors clarify how the analysis was carried out?

- L96 is it Emile-Geay et al., 2008 the first study to proposed the "thermostate mechanism" as driver of El Nino-like response following tropical volcanic eruptions?

- LL96-99 Please, consider rewriting it (not clear what the underlying thermocline does).

- Please make sure the names in the reference are correct: For instance, in #12 Kre?uger

References:

Anchukaitis, K.J. et al. (2010), Influence of volcanic eruptions on the climate of the Asian monsoon region, *Geophys. Res. Lett.*, 37, L22703, doi:10.1029/2010GL044843.

D'Arrigo, R., R. Wilson, and A. Tudhope (2008), The impact of volcanic forcing on tropical temperatures during the past four centuries, *Nat. Geosci.*, 2(1), 51-56, doi:10.1038/ngeo393.

Pausata, F. S. R., L. Chafik, R. Caballero, and D. S. Battisti (2015), Impacts of high-latitude volcanic eruptions on ENSO and AMOC, *Proc. Natl. Acad. Sci.*, 201509153, doi:10.1073/pnas.1509153112.

Stevenson, S., B. Otto-Bliesner, J. Fasullo, and E. Brady (2016), "El Niño Like" Hydroclimate Responses to Last Millennium Volcanic Eruptions, *J. Clim.*, 29(8), 2907-2921, doi:10.1175/JCLI-D-15-0239.1.

Reviewer #2 (Remarks to the Author):

A. Synopsis/summary of key results

This study presents new results from climate observations and simulations (including simulations from CMIP5 and sensitivity experiments carried out for this study) that show, consistently for the first time for both simulations and observations, that strong tropical volcanic eruptions can lead to

an El Niño event within the first two years following a major eruption. As a dynamical interpretation, the authors propose that a key component for triggering such volcanically-driven El Niños is the differential land-sea cooling between the Maritime Continent and the surrounding ocean.

B. Originality and significance

Strong volcanic eruptions is among the major natural forcing factors affecting climate variability on a broad range of timescales. However, there are still large gaps of knowledge about how the climate responds dynamically to volcanic forcing, including whether or not this includes a specific ENSO response. As mentioned in the manuscript, there is no robust indication from climate models and observations about whether an El Niño typically occurs following major eruptions, although we could indeed expect so based on simple theoretical arguments.

In this context, I find the study potentially important and original for two reasons: the attempt to reconcile observations and simulations (achieved mainly by looking at relative SSTs), and the novel dynamical interpretation ascribing a key role to a "volcanic breeze" in the western Pacific warm pool region.

C.- F. Data & methodology, statistics and uncertainties, Conclusions, Suggested improvements

I have a few minor concerns and requests for clarification about the modeling part and interpretation of results. My comments below do not affect my general impression about the good quality of this study.

First of all, the authors use Niño indices to characterize El Niños (mostly Niño3.4 or Niño3, depending on the analysis). There is an ongoing discussion in the scientific literature about different "flavors" of ENSO (see for instance Pascolini et al., 2012 and Lai et al., 2015, and references therein) that are not fully captured by these simple indices. In particular, Pascolini et al. (2012) classified the 1983/84 and 1991/92 El Niños as "Central Pacific" (CP) events. Since CP and "classical" events have different dynamics and climatic implications, it may be worth taking this into account. If so, it would be also useful to check that the employed models produce ENSO variability that include these different types of El Niños. This could be done along with what already reported in the supplemental information. To this regard, a deeper discussion on how the wrong seasonality of ENSO in the IPSL model may affect the results and the conclusions seems to be necessary.

The authors already discuss warming in the Central Pacific (for instance in line 142), so that bringing this aspect of ENSO variability should enrich the paper.

I found the dynamical interpretation overall convincing. Nonetheless there are a few aspects that are not fully clear to me. For instance, I see in Figure 2 that there are noticeable differences between the IPSL and the CNRM model. There are strong and significant anomalies in the eastern Pacific SSTs and zonal wind stress (panels b and d) about at the same time of the Central Pacific warming, that seem to propagate westward. As these are statistically significant anomalies, I think they should be discussed in the light of the proposed mechanism (also in the light of the fact that IPSL is used as basis for the mechanistic experiments).

I have similar requests about the El Niños characteristics and the consistency of the response simulated by the two models in Figure 3. I think it is important to keep the two model results separated also in this figure, as done in the others. This should better clarify, for instance, that the different peaks in the different panels indeed represent successive events in the same model (which could be therefore important for the dynamical interpretation) rather than simply the result of averaging across many realizations from different models. Overall, I think that the statement in lines 92-95 is very important and should be clearly and robustly supported by the associated Figure 3.

Concerning significance, I wonder whether a one-tailed test could be more appropriate for some of

the analyses since you are specifically testing the occurrence of El Niños.

Concerning the hierarchy of model setups for the dynamical understanding of the response, I was wandering whether the authors could provide a bit more details about which effects on ENSO or its precursors the ATM experiment should capture. I wonder whether we should expect any significant signature, if most dynamics in the lower troposphere are substantially damped by the restoring of the boundary layer albedo and of SSTs by design. As far as I understand this, the ATM and LAND experiments are not exactly complementary. Maybe it is worth including a couple of additional words about how the two experiment compare, including how the implemented volcanically forced albedo changes compare. Concerning more specifically the LAND experiment, as far as I envisage it, the wind response may include also effects of larger scale continental cooling over Asia, not only over the Indonesian Maritime continent. Maybe a few more words on this could be helpful.

Minor remarks:

Line 62: a bit more explanation on the reference value of 0.75 would be worth. Also, I wonder whether the authors thought about checking the different models' contribution to the result. In fact, some models may be overrepresented due to their larger ensemble compared to others (looking at Table S1). As a side note, the literature has some papers discussing about whether we should have a climate model democracy or not (for instance, Knutti R., *Climatic Change* (2010) 102:395-404). For instance, weighting of models may be done based on how well they represent ENSO and/or on the severity of their model biases. Possibly, taking the quality of different models into account would provide a stronger model evaluation and therefore strengthen the conclusion from this part of the analysis.

Line 69: as a side note, Zanchettin et al., 2012 looked at stronger eruptions than Pinatubo. Maybe this has some implications for the diagnosed ENSO response. However, this study clarifies that the negative post-eruption anomalies we detected might reflect a change in the background state of the tropical Pacific rather than a La Nina-type response.

Line 93 and Figure 3c: I wonder why the authors write that volcanic forcing favor neutral states instead of La Niñas about Fig 3c. When looking at the figure it seems to me that the peak value in panel c corresponds more or less to 1 std, more than the unperturbed events labeled as positive ENSO (panel a). Maybe this is due to the normalization? If they agree that El Niños would be favored under negative ENSO phases, this would strengthen even more their statement in line 93-95.

G: References

I think the authors account for the most relevant papers on the topic available in the literature. I suggested a couple of references in case the authors decide to perform more in depth analysis of the different types of El Niño:

Pascolini-Campbell, M., D. Zanchettin, O. Bothe, C. Timmreck, D. Matei, J. H. Jungclaus, and H.-F. Graf (2014) Toward a record of Central Pacific El Niño events since 1880. *Theor. Appl. Climatol.*, doi:10.1007/s00704-014-1114-2

Lai, A. W.-C., M. Herzog, and H.-F. Graf (2015) Two key parameters for the El Niño continuum: zonal wind anomalies and Western Pacific subsurface potential temperature. *Clim. Dyn.* 45(11): 3461-3480

H. Clarity and context

The writing and the presentation of the results are clear, and the paper is well structured.

Some minors are listed below:

Line 46: "anomaly in THE Nino3.4 region"

Line 74: maybe change "during" with "starting in"?

Line 80: please check the acronym is defined also in the main text

Line 92: "El NiñoS underway"

Figure 1 caption, line 3: I think "two thirds of the individual events" is unclear. Does that correspond to 3? If so, maybe use that instead.

Figure 2 caption, line 3: dotted \diamond dashed?

Figure 2 caption, line 4: "zonal wind STRESS"?

SI, page 4, line 3: I guess it's "alfa star" in brackets.

Reviewer #3 (Remarks to the Author):

General comments:

The study by Khodri et al. investigates whether explosive tropical volcanic eruptions can trigger El Niño events and what the underlying mechanism might be. The authors mainly rely on climate models: the CMIP5 archive, but also targeted sensitivity experiments run in fully-coupled mode (2 models) as well as in atmosphere-only mode (1 model). With that they go a step further than most other studies on this topic, which just analyze CMIP5 output. The authors conclude that the volcanic forcing sets up a temperature and pressure gradient across the land-sea border of the Eastern Pacific maritime continent and the Pacific ocean, which in turn creates westerly wind anomalies that then favor the development of El Niño.

The proposed mechanism is interesting and the sensitivity experiments add great value to the discussion of this mechanism. I see, however, a few major issues with the potential robustness of the results (mainly related to lack of support from observations), which might prohibit this study to be more than an incremental one. In that, I am not sure it meets the "Nature standard", but I leave this to the editor to decide. The study is definitely worth publishing somewhere, although a few major comments need to be addressed beforehand. I will try to lay out some ideas that might be helpful.

Major comments:

- The largest issues with this paper is its little observational evidence. It is essentially a nice modeling paper, motivated by model results, while the motivation from observations is small (and maybe not robust?). Take Figure 1a and 1b, which is really the only significant effort by the authors to root their hypothesis in observations: Out of the 5 volcanic eruptions, 1 does not show the response, 2 had an El Niño underway already. The relative SST anomalies are considered significant when 2/3 of the eruptions show the same sign. I suppose that means when 4 out of 5 eruptions have the same sign (please clarify, the caption is ambiguous). But 2 of those 4 were going to have an El Niño anyways. While

acknowledging the tremendous difficulties to find sufficient observational evidence for such rare events like volcanoes, the “significance test” provided here is by no means sufficient. This needs to be discussed more. Also, I would like to see the CMIP5 Hovmoller diagram next to the observed one. Please make a panel (c) in Figure 1, that is the same as panel (b), but for CMIP5. The authors mention in one small sentence at the end of the paper that the models and observations do not agree on the timing of the El Nino after the eruptions. This is not illustrated and discussed and leaves the impression that the authors want to sweep this under the rug. This is important, because it highlights the challenge of comparing the sparse observations with the many (and seemingly robust) model results. I do not think anyone expects this paper to solve all the problems around this topic (exactly because it is so hard to prove in observations), so an open discussion of the caveats will benefit the reader.

- Along the same lines as above, in Figure 2 the authors should provide a third row with the same results from observations. Since that figure deals with Pinatubo, there should be sufficient observations available (at least that is stated as the authors’ motivation to focus on Pinatubo).
- There is a number of studies and model datasets available that could help to resolve a bit more the potential convolution of volcanic eruptions during which El Nino is already underway. Froelicher et al. (2013, GBC) ran a very similar setup as the coupled sensitivity experiments presented here with a version of the NCAR model, also targeting Pinatubo. The focus of that paper is on the carbon cycle, so maybe their model results can be revisited with respect to the issue here. More recently, Douville et al. (2015, GRL) and Lehner et al. (2016, GRL) used so-called POGA experiments with CNRM (Douville) and CESM and GFDL (Lehner). These experiments are fully coupled, but the eastern tropical SSTs are nudged towards observations, so the ENSO evolution is as observed. Contrasting these simulations with the historical runs of the respective model (which the authors already use here in their CMIP5 analysis) could inform how much of the observed response (e.g., Figure 1 here) is due to the El Nino-volcano coincidence. In fact, that is the main pitch of the Lehner et al. paper, at least with respect to global temperature
- I like the idea of relative SSTs telling us more than raw SST anomalies about the dynamic coupling of atmosphere and ocean, but the notion that they are co-located with precipitation anomalies does not seem to hold in IPSL (Figure 2b and 2c). Again, observations would be helpful to put the modeling results into perspective. If there is a model bias in IPSL regarding the precipitation response to radiative or dynamic forcing in the tropical Pacific, then I would expect that bias to impact the installment of the wind response, that is claimed to be the main trigger for the El Nino after volcanoes (as suggested by the LAND experiment). Why did the authors choose to do these atmosphere-only experiments with IPSL and not CNRM, which has more of a CMIP5-type coupling of relative SSTs and precipitation (see Figure S2)?
- The role of the linear ocean model is not explained sufficiently. Since the authors prescribe SSTs in the atmosphere-only experiment “OCEAN”, what does the SSTs derived from the linear model tell us?

Minor comments (no particular order):

- There are no numbers given on the relative likelihood of El Nino after volcanoes. How much more likely are they according to this new definition of El Nino with the relative SST anomalies? See for example Maher et al. (2015, GRL).
- L91: maybe say “relationship” rather than “cycle”.
- A few things about Figure 3a: why do the sensitivity experiments with Pinatubo (pink) start systematically higher than the control (black)? Also, the reverse seems to be the case in Figure 3c. Is this due to the sampling from the historical simulations? At least in Figure 3a, taking into account this offset would bring the pink and black lines into agreement, meaning the volcanic forcing has no discernible effect on those cases. Please clarify. Also, are the authors worried about differences in the response due to sampling from different background climates if they sample across the whole range of the historical simulations? Why is the confidence interval narrower for panel (c)? Is it because there are more samples for ENSO neutral? Please clarify. Along those lines, it would be good to also show the full range, not just the 90% interval, because then the authors could add the observations in Figure 3, allowing for an apples-to-apples comparison. The lines seem quite smooth; was there a filtering applied?
- L100: delete “rather”.
- A suggestion: reorder text and figures so that current Figure 3 comes before Figure 2; currently, the text jumps back and forth between those two figures and it is unnecessarily hard to follow.
- L110: what is meant exactly with “overall response”?
- Paragraph starting L112: consider providing at least a bit more details on the atmosphere-only experiments here; currently it is not comprehensible what you did. The supplementary is good. Consider adding a simple table along these lines:

Experiment name	Origin of ocean forcing	Origin of land forcing	Atmosphere forcing
Control	Control	Control	No volcanoes
ATM	Control	Control	Volcanoes
LAND	Control	Forced	No volcanoes
OCEAN	Forced	Control	No volcanoes
ALL	Forced	Forced	Volcanoes

- L134: “...stage that is prescribed via the SST anomalies...”
- L139: “the western Pacific”
- L210-212: please clean up the notation of the experiments; at one point the authors talk about “CGCM control ensemble”, at one point about “IPSL-CM5A ensemble”. As mentioned earlier, the paper needs to discuss in more detail the implications of the fact that the main sensitivity experiments are based on just one model.
- Figure 4: please add the results from the fully-coupled simulations as well, both for the time series and the maps. Also, add a series of maps for the observations.
- As mentioned earlier, Figure 4b is a bit confusing, since the SST anomaly in the OCEAN experiment is not the same as in the fully-coupled experiment, even though that is where the SST anomalies have been taken from to prescribe them to the atmosphere-only model

setup. Essentially the same is valid in Figure S4b, where there is a clear discrepancy between the ALL results and the OAGCM early on in the simulation.

Reviewer #1 (Remarks to the Author):

The authors investigate the ENSO's response to tropical volcanic eruptions both (i) using CMIP5 models, and (ii) performing a set of sensitivity simulations using two different climate models (IPSL-CN5A-LR and CNRM-CM5). The authors show that the relative SST (RSST) - instead of SST - should be used to detect the El Niño-like response after volcanic eruptions in CMIP5 model. The authors then point to westerly wind anomalies in the fall following the eruption, as triggers for El Niño-like response. They claim that the wind anomalies are caused by a giant sea breeze generated by the stronger cooling of the Maritime Continent compared to the equatorial Pacific following a tropical eruption.

The results are potentially of interest for the Nat. Comm. broad audience. However, I have several concerns that prevent me to recommend it for publication in its present form.

Major comments:

1) THE MODEL ADOPTED:

a. The authors used two climate models to test their hypothesis. The models used are IPSL-CN5A-LR and CNRM-CM5. The authors highlight in the supplementary material that the IPSL model has some issues with the ENSO seasonality. Hence, IPSL-CN5A-LR is a concern and should not be used for studies investigating changes in ENSO. Not only IPSL-CN5A-LR has an ENSO seasonality peaking in the wrong season, but also all the feedbacks associated with the ENSO dynamics are largely off (see Fig. 3 in Bellenger et al., 2014 - IPSL (a)). It is surprising that the authors did not adopt the IPSL-CN5B-LR model instead (see IPSL (c) in Fig. 3 in Bellenger et al., 2014), which seems to perform better. On the other hand, the CNRM model seems to have a good representation of ENSO mean climate and feedbacks.

Therefore, I would not recommend to use IPSL-CN5A-LR, but instead, to switch to IPSL-CN5B-LR.

For instance, figure 2 clearly shows a notable different evolution in the SST anomalies following the volcanic eruptions, which may simply be a result of the wrong ENSO representation in the IPSL model.

b. The authors opted to use IPSL to CNRM to perform the uncoupled sensitivity simulations; however, no justification is presented regarding this choice. I am puzzled about this choice, since in the sensitivity experiments the authors need to prescribe the SST, which has a wrong seasonality in the Tropical Pacific.

2) THE APPROACH:

a. It is not clear to me how the authors created the Pinatubo and unforced ensemble members. The authors state that the unforced members "were performed, only differing from the forced ensemble by the specified Pinatubo volcanic forcing." In that case, the first 5 months (Jan-May) should be identical in the two ensembles (forced and unforced) since the eruption occurs in June. However, Figure 3 clearly shows that they do not have the same initial conditions. Even if a smoothing (3 or 5 month running mean) is applied

to the Nino3 index at least the first 3-4 months should be identical or barely distinguishable, which is clearly not the case.

In case, the authors - for unknown reasons - have used slightly different initial conditions in January, in order to initialize the unforced ensemble members relative to the volcano members, the simulations should be re-run: the benefit of running your own simulations compared to CMIP is because you can clearly disentangle the role of volcanic eruption performing volcano and unforced run starting from identical initial conditions.

b. The authors blend together the results from the two models in Figure 3, which is quite misleading. This is particularly important, as in one of the models the ENSO seasonality is completely off, peaking in spring/summer. The authors need to keep the models' results separated if they will still use IPSL-CN5A-LR.

We agree with the reviewer's concerns raised above. To address each issues we completely revised our experimental protocol and proceeded as follow:

- We now use the IPSL-CM5B model, which has a better ENSO seasonality and a nice normalized Pacific Warm Water Volume-N3.4, 6 months lagged correlation (see Fig. 2 and Fig. 3 in Jourdain et al 2016).
- We have improved the experimental strategy as suggested by the reviewer and as follows. 30-member ensembles (applying small perturbations to SST to generate the ensemble) are now started in June, just before the eruption. To evaluate the effect of the ENSO state on the response to volcanism, we have carefully chosen 3 initial states in the IPSL-CM5B CMIP5 historical simulation (Dufresne et al, 2013) based on the normalized Pacific WWV in May (e.g., Meinen and McPhaden, 2000) that cover the main 3 phases of ENSO cycle (discharged, near neutral, recharged, see Supplementary Fig.S8).
- We agree that it would have been ideal to perform the whole set of sensitivity experiments using both IPSL and CNRM models. These experiments however involve heavy technical developments and computing resources, which are not available at CERFACS (where CNRM model is run) at this time.

The revised manuscript thus now focuses on IPSL-CM5B new ensembles, while the results from the CNRM model 23-members ensemble simulations are shown as Supplementary material (see Fig.S6). The new experimental protocol is now described in Section 3 of the Supplemental Information. We think that it addresses the reviewers concerns, and that it is much more relevant for our purpose. We are grateful to the reviewer for these suggestions. These new simulations allow addressing the main reviewer's concerns regarding ENSO seasonality and feedbacks.

3) THE MECHANISM:

The authors show that El Nino response starts with a wind stress anomaly in October. They perform a set of sensitivity studies to try to understand the cause of such wind stress anomaly in the western tropical Pacific. The authors conclude that the LAND experiment (i.e. when they just change the surface temperature over land) is able to reproduce the westerly wind anomaly in the model. The authors explain the mechanism

based on relatively local (regional) scale changes, without investigating large-scale changes that could have triggered it. **I am skeptical that the cooling (which is barely significant and of the order of 0.2{degree sign}C) of the Maritime Continent alone could trigger the chain of events described by the authors.** They show the change in temperature over the Maritime Continent (no specification of exact lon and lat used) and Western Equatorial Pacific SST in Figure S5; however, the western boundary of the Western Equatorial Pacific domain considered by the authors is 90 {degree sign}E, i.e. west of the entire Maritime continent). Since the authors are claiming that the difference in temperature is causing the giant breeze, the two areas (land vs. ocean) should therefore not overlap and they should consider a further eastern domain (e.g. 120 - 180{degree sign}E).

On the other hand, a recent study (**Pausata et al., 2015**) showed that **the NH cooling would lead to a shift south of the ITCZ in the Pacific**, which would weaken the trades. They also showed that for an eruption occurring in summer, the surface wind changes and the ITCZ shift over the Pacific would occur around September/October, because of lower response over the Pacific Ocean relative to the Asian continent were the shift already started in the summer with a weakening of **the Asian Monsoon system (AMS)**. They simulated high-latitude volcanic eruptions; nevertheless as the authors themselves pointed out the land will cool more rapidly than the ocean, i.e. the NH is likely to cool more than the SH. Hence, the same mechanism can be at play in your simulations. The weakening of the AMS and/or the ITCZ shift over the Pacific could be the cause of the westerly anomalies also in your case. **Recently, Stevenson et al. (2016) also mentioned the ITCZ shift as trigger of the El Nino-like response.**

The authors should show and quantify the cooling as well as potential shift in the ITCZ and discuss their results in the context of previous studies (Pausata et al., 2015 and Stevenson et al., 2016). Hence a map showing the global cooling and precipitation changes in different seasons (JJA, SON, DJF) would be beneficial.

The larger landmass in the Northern Hemisphere could indeed induce a southward shift of the ITCZ in boreal summer and fall, weaken the AMS and cause the westerly anomalies, as for the extratropical NH eruptions in Pausata et al. (2015) and Stevenson et al. 2016. However, as shown in Fig.R1 below, this is not the case in both CNRM and IPSL model ensembles. The Pinatubo aerosol cloud is still mostly confined within the tropics during both seasons, resulting in a rather symmetrical cooling across the tropics. Precipitation and wind anomalies in JJA and SON 1991 do not reveal any southward shift of the ITCZ or a weakening of the AMS that could explain the westerly wind anomaly in Western Pacific in summer-fall 1991. Surface temperature and precipitation anomalies of the CMIP5 models in summer and fall are consistent with results from both IPSL and CNRM model ensembles (not shown).

Fig. R1. IPSL-CM5B 30 members (neutral case) paired anomalies for surface temperatures, mean precipitation and surface winds (red vectors) in June-July-August (JJA) and September-October-November (SON). Only significant wind anomalies are plotted and the white contours indicate statistically significant anomalies at the 90% confidence level, based on a two-tailed Welch’s t -test.

To rule out which kind of mechanism is driving the El Niño-like response, an additional sensitivity test is required, in which the cooling is applied only to Southeast Asia.

We are thankful for the reviewer’s suggestion and to the editor for giving us the needed time to address this point and develop such experiments. These new experiments have allowed us to identify the key mechanisms, which are interestingly different from those suggested by earlier studies, and overall the results yield richer and more physically robust results regarding how explosive volcanisms can induce El Niño-like events.

The revised manuscript now uses the following suite of 30 members (forced and unforced) sensitivity experiments, performed with the atmospheric components of the IPSL-CM5B model. The set of sensitivity experiments are based on the coupled model ensemble starting from Neutral initial conditions:

Experiment Name	Origin of ocean boundary condition	Origin of land boundary condition	External forcing in the atmosphere
Control	Control	Control	No volcano
ALL	Pinatubo	Pinatubo	Volcano
OCEAN	Pinatubo	Control	No volcano
ATM	Control	Control	Volcano
LAND	Control	Pinatubo	No volcano
LAND-T	Control	Pinatubo 25°N-25°N and control elsewhere	No volcano
LAND-ET	Control	Pinatubo 25°N-90°N; 90°S-25°S and control elsewhere	No volcano
LAND-SEA	Control	Pinatubo 90°E-160°W; 10°N-30°N and control elsewhere	No volcano
LAND-MC	Control	Pinatubo 90°E-150°W; 10°S-10°N and control elsewhere	No volcano
LAND-Africa	Control	Pinatubo 15°W-50°E; 10°S-30°N and control elsewhere	No volcano

The new experimental protocol is detailed in Section 3.2.1 of the supplementary information. As shown on Figure S10, the favourable comparison of the new ALL ensemble mean with the coupled model mean precipitation, surface temperature and surface wind anomalies suggests that our method is sound.

The new sensitivity experiments allow exploring the land cooling only over the tropics (LAND-T), the extra tropics (LAND-ET), Africa (LAND-Africa), South East Asia (LAND-SEA) and the Maritime continent (LAND-MC) only. As discussed before, based on Fig.R1, the land cooling over South East Asia or extra tropics only does not induce any significant precipitation and windstress anomalies over the western equatorial Pacific in fall (Fig. S12). The set of sensitivity experiments rather shows that it is the tropical Africa land cooling which dominates at the start, with significant LCS SST anomalies in Niño3 from August to November (cf. new Fig. 3h). This suggests that the mechanism hypothesized by Ohba et al. 2013, and which we were also suggesting in the first version of this manuscript, is not the dominant one. We would like to thank the reviewer for having advised us to do further sensitivity experiments to test this mechanism.

Rather than the MC/South East Asia or extratropics, it is hence the cooling over the largest tropical land mass, namely Africa, that has the strongest effect. Decreased precipitation and tropospheric diabatic heating over tropical Africa forces zonal surface wind anomalies consistent with equatorial Kelvin waves following a Matsuno/Gill-type

response (Gill, 1980). The Kelvin waves are emitted to the east of Africa with SLP/westerly anomalies, which reach the western Pacific in fall 1991 (qualitatively as in the Matsuno/Gill response to a tropospheric heating anomaly, with similar physical mechanisms to some highlighted for a distinct scientific question by Li et al. 2015).

Our results show that the tropical African cooling explains 80% of the initial wind anomalies while the climatological position of the ITCZ straddling the equator in summer and fall provides the energy source for the propagation of the wave towards Western Pacific (See Fig. 4; Fig. S13a-c). Our results therefore confirm the secondary role played by the zonal equatorial Pacific SST gradient while allowing us narrowing down the continental region at the origin of the initial westerly wind anomaly.

We have now modified the text to describe these new results.

Finally, although I'm not familiar with the method used by the authors for the uncoupled sensitivity test (albedo change), the authors should show, for instance, that the cooling simulated over land is comparable, both in the LAND experiment and in the fully coupled experiment.

We have now detailed more the method in section 3.2.1 of the revised Supplemental Information and show on Figure S10 that our method allows reproduce the cooling over land and ocean in the LAND and OCEAN dedicated experiments.

Minor comments:

- The authors mentioned that the proxy archives suggest the occurrence of El Niño-like conditions following volcanic eruptions. However, to my knowledge, this situation has not yet been fully clarified using proxies. For instance, D'Arrigo et al., 2009 and Anchukaitis et al., 2010 showed a La Niña like response after volcanic eruptions. Therefore, based on previously published studies, I would be cautious stating that proxy records indicate El Niño conditions after volcanic eruptions.

Our argument is sustained by recent studies (Li et al 2013, Stevenson et al 2017) that we now cite in the text. But we agree that the debate among proxy reconstructions of El Niño conditions after volcanic eruptions would require further analyses and a proper discussion. We have rephrased the sentence so that it is less assertive. See line 50-52 in the main manuscript.

- Given the broad NCOMMS audience I would suggest to use the term Southeast Asia rather than Maritime Continent that is less known (or specify the geographical region).

Since we now investigate the influence from Southeast Asia and Maritime Continent separately we decided to keep this term (see Fig. S12). We however now specify the geographical region for both Southeast Asia and Maritime Continent in Table S2.

- LL 47: the authors state that the RSST anomaly in Niño3.4 region reaches 1.1°C , that has a 0.3% probability as shown in Figure S1. However, if I understood correctly Figure S1, this figure shows the pdf of averaged NDJ RSST anomalies, while the 1.1°C anomaly refers to just one month. The 3-month average should

probably be around $0.9/0.95^{\circ}\text{C}$; hence, the authors should correct the probability ($\sim 0.7\text{-}1\%$).

We apologize for the earlier Fig. S1 caption, which was not clear enough. The 1.1°C anomaly was for the NDJ 3 months average, averaged over the 5 eruptions years. And the PDF was constructed consistently, randomly picking 5 different years among the historical 1870-2010 period, and then averaging their 3 months-averaged NDJ RSST anomalies. So we confirm that it is a 0.3% probability. We have now clarified the caption accordingly.

By the way, it is quite surprising that the largest eruption considered in this study does not produce a warm anomaly. Related to that I question the choice of using 1961-1990 as climatology! What is the simulated trend in SST from 1880 to 2000? I would expect $0.3/0.4^{\circ}\text{C}$ warmer SST in the tropical Pacific (NINO region) in the 1961-1990 compared to the decade of the Krakatau eruption 1880-1889.

We actually tested several methods to compute the anomalies: using 1961-1990 as climatology, the preceding 5 years, 10 years, or the whole period. The results are marginally affected by the baseline period used to compute the anomalies. We decided to use SST anomalies computed relative to the 5-year mean preceding each eruption (to remove multidecadal variability and long-term trends) and smoothed with a three-month Hanning filter (to filter out intraseasonal variability), as often done in volcanic studies. The weak signal for the Krakatau eruption could be due to the ocean preconditioning (as demonstrated by our results), to the late timing of the eruption, which may explain that the El Niño signal seems to be stronger the 2nd year, and/or to the scarceness in observations at the beginning of the 19th century. The earliest part of the record (late 19th century) is, indeed, less reliable due to lower spatial resolution.

- The authors show that RSST is better than SST in capturing the El Niño response in CMIP5 models. However, I would be very curious to see the same plot in figure 1a and b done using SST instead of RSST. This figure should go in the supplementary to support that statement.

We have included the figure in the supplementary in Section 1 (see Fig. S1). The difference between raw (SI Fig. S1) and relative SST (main Fig.1) in observations is weak, with a slight increase in amplitude and significance (less spread among eruptions). This difference is much larger in CMIP5 models (see Fig. S3a,b). This result is not surprising as the largest uncertainties in the estimates of radiative forcing from CMIP5 historical simulations occur during periods of volcanic activity (Santer et al. 2014) and generally models tend to overestimate the observed post-eruption global surface cooling (Marotzke and Forster 2015). Relying on relative SST and precipitations anomalies therefore helps correct this bias and help reconcile models and observations as it reveals the dynamical ENSO response to volcanic forcing in CMIP5 models.

- Fig. 1: In panel a, I assume 0.6°C should actually be 0.8°C . Please correct it or use a linear scale.

We have corrected this in the revised Fig.1.

- L62: 0.75 what? Standard deviation? Is this the common metric used when analyzing CMIP5 models? In observation the 0.4{degree sign}C threshold (exceeded for at least 6 consecutive months) is used for Niño3.4 which is 0.5 standard deviation.

We now use 0.5 standard deviation in Niño3.4 index to detect El Niño events (it does not change the main results). Fig.1c has been updated accordingly.

- LL64-67 and Fig. 1c: I must stay I haven't followed exactly how that analysis has been performed. Could the authors clarify how the analysis was carried out?

We have now included more details on how the analyses were performed in Section 2 of the Supplemental Information and rephrased the corresponding sentences in main manuscript to clarify this (L61-L67).

- L96 is it Emile-Geay et al., 2008 the first study to proposed the "thermostate mechanism" as driver of El Nino-like response following tropical volcanic eruptions?

We have corrected this and now also cite Adams et al, 2003 in addition to Emile-Geay et al., 2008.

- LL96-99 Please, consider rewriting it (not clear what the underlying thermocline does).

We have rephrased the corresponding sentence (L90-L93).

- Please make sure the names in the reference are correct: For instance, in #12 Kre?uger

Done

References:

- Anchukaitis, K.J. et al. (2010), Influence of volcanic eruptions on the climate of the Asian monsoon region, *Geophys. Res. Lett.*, 37, L22703, doi:10.1029/2010GL044843.
- D'Arrigo, R., R. Wilson, and A. Tudhope (2008), The impact of volcanic forcing on tropical temperatures during the past four centuries, *Nat. Geosci.*, 2(1), 51-56, doi:10.1038/ngeo393.
- Pausata, F. S. R., L. Chafik, R. Caballero, and D. S. Battisti (2015), Impacts of high-latitude volcanic eruptions on ENSO and AMOC, *Proc. Natl. Acad. Sci.*, 201509153, doi:10.1073/pnas.1509153112.
- Stevenson, S., B. Otto-Bliesner, J. Fasullo, and E. Brady (2016), "El Niño Like" Hydroclimate Responses to Last Millennium Volcanic Eruptions, *J. Clim.*, 29(8), 2907-2921, doi:10.1175/JCLI-D-15-0239.1.

Reviewer #2 (Remarks to the Author):

A. Synopsis/summary of key results

This study presents new results from climate observations and simulations (including simulations from CMIP5 and sensitivity experiments carried out for this study) that show, consistently for the first time for both simulations and observations, that strong tropical volcanic eruptions can lead to an El Niño event within the first two years following a major eruption. As a dynamical interpretation, the authors propose that a key component for triggering such volcanically-driven El Niños is the differential land-sea cooling between the Maritime Continent and the surrounding ocean.

B. Originality and significance

Strong volcanic eruptions is among the major natural forcing factors affecting climate variability on a broad range of timescales. However, there are still large gaps of knowledge about how the climate responds dynamically to volcanic forcing, including whether or not this includes a specific ENSO response. As mentioned in the manuscript, there is no robust indication from climate models and observations about whether an El Niño typically occurs following major eruptions, although we could indeed expect so based on simple theoretical arguments.

In this context, I find the study potentially important and original for two reasons: the attempt to reconcile observations and simulations (achieved mainly by looking at relative SSTs), and the novel dynamical interpretation ascribing a key role to a "volcanic breeze" in the western Pacific warm pool region.

C.- F. Data & methodology, statistics and uncertainties, Conclusions, Suggested improvements

I have a few minor concerns and requests for clarification about the modeling part and interpretation of results. My comments below do not affect my general impression about the good quality of this study.

First of all, the authors use Nino indices to characterize El Niños (mostly Nino3.4 or Nino3, depending on the analysis). There is an ongoing discussion in the scientific literature about different "flavors" of ENSO (see for instance Pascolini et al., 2012 and Lai et al., 2015, and references therein) that are not fully captured by these simple indices. In particular, Pascolini et al. (2012) classified the 1983/84 and 1991/92 El Niños as "Central Pacific" (CP) events. Since CP and "classical" events have different dynamics and climatic implications, it may be worth taking this into account. If so, it would be also useful to check that the employed models produce ENSO variability that include these different types of El Niños. This could be done along with what already reported in the supplemental information. To this regard, a deeper discussion on how the wrong seasonality of ENSO in the IPSL model may affect the results and the conclusions seems to be necessary.

The authors already discuss warming in the Central Pacific (for instance in line 142), so that bringing this aspect of ENSO variability should enrich the paper.

I found the dynamical interpretation overall convincing. Nonetheless there are a few aspects that are not fully clear to me. For instance, I see in Figure 2 that there are noticeable differences between the IPSL and the CNRM model. There are strong and significant anomalies in the eastern Pacific SSTs and zonal wind stress (panels b and d) about at the same time of the Central Pacific warming, that seem to propagate westward. As these are statistically significant anomalies, I think they should be discussed in the light of the proposed mechanism (also in the light of the fact that IPSL is used as basis for the mechanistic experiments).

We agree that the wrong seasonality of the IPSL-CM5A model was a limitation to the robustness of our results. As explained in the response to the first reviewer we addressed this issue by revising our experimental protocol (please see the response to the first reviewer) and now use the IPSL-CM5B model. In addition, we now have a clean experimental setup to study the interaction of the volcanic forcing and natural ENSO cycle in the IPSL-CM5A model.

It is delicate to study details of the timing of the El Niño response to volcanism in the CNRM and CMIP5 ensembles. These ensembles indeed start from a random mix of oceanic initial conditions as far as the ENSO cycle is concerned. This is not directly comparable to our new (cleaner) protocol, which uses WWV preconditioning to explore the response of the main 3 phases of ENSO cycle (discharged, near neutral, recharged). For this reason, the CNRM experiments have been moved to the SI for illustration, and the paper now focuses on results obtained with the IPSL-CM5B model.

Nevertheless, the points raised by the reviewer (anomalies in the eastern Pacific SSTs and zonal wind stress, central Pacific Warming) are now addressed in the new version of the manuscript (see L105-113, Fig 3 and Fig. 4 schematic for a summary). There is indeed a first central Pacific Warming as a response to the first westerly wind in fall of the eruption year. This first warming is damped during the beginning of the following year (probably by negative delayed feedback which usually terminate an El Niño). The Bjerkness feedback and continued effect of land cooling on Pacific winds then lead to El Niño-like anomalies during the year after the eruption.

I have similar requests about the El Niños characteristics and the consistency of the response simulated by the two models in Figure 3. I think it is important to keep the two model results separated also in this figure, as done in the others. This should better clarify, for instance, that the different peaks in the different panels indeed represent successive events in the same model (which could be therefore important for the dynamical interpretation) rather than simply the result of averaging across many realizations from different models. Overall, I think that the statement in lines 92-95 is very important and should be clearly and robustly supported by the associated Figure 3.

We have completely revised our protocol and now the impact of the eruption on the 3

main phases of ENSO cycle is addressed with 3 IPSL-CM5B model 30 members ensembles (one starting from “discharged”, one from neutral and one from “recharged” equatorial Pacific heat content). We had not the resources to develop the same set of experiments with the CNRM model (see response to the first reviewer). We however think that this aspect is robust and consistent with recently published studies relying on other models (e.g. Predybaylo et al, 2017). Our revised setup clearly shows that volcanic forcing tends to shorten La Niñas (Fig. 2a), lengthen El Niños already underway (Fig. 2b) and favours El Niño-like events instead of neutral states (Fig. 2c), and partly consistent with Ohba et al. (2013). This gives us confidence in the robustness of our results and therefore allows us focus on the physical mechanism at play on the tier-two set of experiments.

Concerning significance, I wonder whether a one-tailed test could be more appropriate for some of the analyses since you are specifically testing the occurrence of El Niños.

We indeed used a one-tailed test and monte carlo approach (see Fig. S2) when testing the occurrence of El Niños and a two-tailed Welch’s *t*-test when analysing paired anomalies (considering the variance of forced and unforced model ensembles) and Student’s *t*-test when analysing anomalies to the climatology.

Concerning the hierarchy of model setups for the dynamical understanding of the response, I was wandering whether the authors could provide a bit more details about which effects on ENSO or its precursors the ATM experiment should capture. I wonder whether we should expect any significant signature, if most dynamics in the lower troposphere are substantially damped by the restoring of the boundary layer albedo and of SSTs by design. As far as I understand this, the ATM and LAND experiments are not exactly complementary. Maybe it is worth including a couple of additional words about how the two experiment compare, including how the implemented volcanically forced albedo changes compare. Concerning more specifically the LAND experiment, as far as I envisage it, the wind response may include also effects of larger scale continental cooling over Asia, not only over the Indonesian Maritime continent. Maybe a few more words on this could be helpful.

We agree with this remark, that the protocol was not described clearly enough. ATM and LAND experiments are actually exactly complementary. The sum of the ATM, LAND and OCEAN experiments is close to results of the coupled model (i.e. internal noise and non-linearities are relatively small in our experimental framework), which is one of the advantages of our method constraining albedo instead of land surface temperature. We have given details on this issue in our response to rev 1 and in the Supplementary Information.

Minor remarks:

Line 62: a bit more explanation on the reference value of 0.75 would be worth. Also, I wonder whether the authors thought about checking the different models' contribution to

the result. In fact, some models may be overrepresented due to their larger ensemble compared to others (looking at Table S1). As a side note, the literature has some papers discussing about whether we should have a climate model democracy or not (for instance, Knutti R., Climatic Change (2010) 102:395-404). For instance, weighting of models may be done based on how well they represent ENSO and/or on the severity of their model biases. Possibly, taking the quality of different models into account would provide a stronger model evaluation and therefore strengthen the conclusion from this part of the analysis.

We agree with the reviewer that, as some models have better ENSO than others, it might be a good idea to rely on CMIP models sub-samples. However, there is interplay between improving the quality of models and decreasing their number .i.e. when using less models, the results are more aliased by internal climate variability. We tested our result by using 4 “good ENSO models” with a correct ENSO seasonality and feedbacks (CNRM-CM5, CSIRO-Mk3-6-0, HadGEM3 and GFDL CM2.1, according to Bellenger et al. 2014), each of which had 10 members ensemble in the CMIP5 database. The results are however qualitatively similar to the whole set of 106 CMIP5 models (not shown). We have decided not to mention it in the original version of this manuscript for the sake of simplicity.

Line 69: as a side note, Zanchettin et al., 2012 looked at stronger eruptions than Pinatubo. Maybe this has some implications for the diagnosed ENSO response. However, this study clarifies that the negative post-eruption anomalies we detected might reflect a change in the background state of the tropical Pacific rather than a La Nina-type response.

It is true that our results (dynamical ENSO response) could also apply to stronger eruptions. We included this citation in the revised manuscript.

Line 93 and Figure 3c: I wonder why the authors write that volcanic forcing favor neutral states instead of La Niñas about Fig 3c. When looking at the figure it seems to me that the peak value in panel c corresponds more or less to 1 std, more than the unperturbed events labeled as positive ENSO (panel a). Maybe this is due to the normalization? If they agree that El Niños would be favored under negative ENSO phases, this would strengthen even more their statement in line 93-95.

We updated Fig 3 (now figure 2) using the new set of simulations, in particular where initial conditions are based on the WWV. It is now clear that that volcanic forcing tends to shorten La Niñas (Fig. 2a), lengthen El Niños already underway (Fig. 2b) and favours El Niño-like events instead of neutral states (Fig. 2c).

G: References

I think the authors account for the most relevant papers on the topic available in the literature. I suggested a couple of references in case the authors decide to perform more in depth analysis of the different types of El Niño:

Pascolini-Campbell, M., D. Zanchettin, O. Bothe, C. Timmreck, D. Matei, J. H. Jungclaus, and H.-F. Graf (2014) Toward a record of Central Pacific El Niño events since 1880. *Theor. Appl. Climatol.*, doi:10.1007/s00704-014-1114-2

Lai, A. W.-C., M. Herzog, and H.-F. Graf (2015) Two key parameters for the El Niño continuum: zonal wind anomalies and Western Pacific subsurface potential temperature. *Clim. Dyn.* 45(11): 3461-3480

Thanks for the suggested citations. We did not perform in depth analyses on the different types of El Niño, so we did not include these citations.

H. Clarity and context

The writing and the presentation of the results are clear, and the paper is well structured.

Some minors are listed below:

Line 46: "anomaly in THE Nino3.4 region"

Line 74: maybe change "during" with "starting in"?

Line 80: please check the acronym is defined also in the main text

Line 92: "El NiñoS underway"

Figure 1 caption, line 3: I think "two thirds of the individual events" is unclear. Does that correspond to 3? If so, maybe use that instead.

Figure 2 caption, line 3: dotted à dashed?

Figure 2 caption, line 4: "zonal wind STRESS"?

SI, page 4, line 3: I guess it's "alfa star" in brackets.

All minor changes above have been implemented.

Reviewer #3 (Remarks to the Author):

General comments

The study by Khodri et al. investigates whether explosive tropical volcanic eruptions can trigger El Niño events and what the underlying mechanism might be. The authors mainly rely on climate models: the CMIP5 archive, but also targeted sensitivity experiments run in fully-coupled mode (2 models) as well as in atmosphere-only mode (1 model). With that they go a step further than most other studies on this topic, which just analyze CMIP5 output. The authors conclude that the volcanic forcing sets up a temperature and pressure gradient across the land-sea border of the Eastern Pacific maritime continent and the Pacific ocean, which in turn creates westerly wind anomalies that then favor the development of El Niño. The proposed mechanism is interesting and the sensitivity experiments add great value to the discussion of this mechanism. I see, however, a few major issues with the potential robustness of the results (mainly related to lack of support from observations), which might prohibit this study to be more than an incremental one. In that, I am not sure it meets the “Nature standard”, but I leave this to the editor to decide. The study is definitely worth publishing somewhere, although a few major comments need to be addressed beforehand. I will try to lay out some ideas that might be helpful.

Major comments:

- The largest issues with this paper is its little observational evidence. It is essentially a nice modeling paper, motivated by model results, while the motivation from observations is small (and maybe not robust?). Take Figure 1a and 1b, which is really the only significant effort by the authors to root their hypothesis in observations: Out of the 5 volcanic eruptions, 1 does not show the response, 2 had an El Niño underway already. The relative SST anomalies are considered significant when 2/3 of the eruptions show the same sign. I suppose that means when 4 out of 5 eruptions have the same sign (please clarify, the caption is ambiguous). But 2 of those 4 were going to have an El Niño anyways. While acknowledging the tremendous difficulties to find sufficient observational evidence for such rare events like volcanoes, the “significance test” provided here is by no means sufficient. This needs to be discussed more.

We agree with the reviewer that finding sufficient observational evidence for such rare volcanic events is a difficult task, and that for some of the eruptions, there was an El Niño underway, including Pinatubo for which there were signs of El Niño onset at the time of the eruption, with a Niño3.4 relative SST anomaly of $\sim 0.6^{\circ}\text{C}$ in May 1991 (and with a favourable WWV conditions in oceanic reanalyses, see new Fig.S7). We also agree with the reviewer that simply counting the number of consistent signs, as in Fig. 1a, is a very crude “significance test”. That is why we had also developed a more complex Monte Carlo significance test approach, which result is given in Supplementary Figure S2. In this figure, we are actually testing the likelihood that the composite of 5

randomly selected years within the historical period has a November-January Niño3.4 relative SST anomaly of 1.1°C, as for the composite of the 5 eruptions in the historical period. This likelihood is very small (0.3%). We now explain this Monte Carlo strategy better in the Figure S2 caption and in Supplemental Information Section 1. In figure 1, we now also only mark points with 5 out of 5 consistent signs in order to make that simple test more severe. Supposing that there is the same probability of having positive and negative signs, according to the binomial law, the likelihood for 5/5 drawing of the same sign is $p=0.03$, i.e. corresponding to a 97% significance level (this is now mentioned in the Supplementary Information section). In any case as the reviewer mentioned, it is very difficult to find sufficient observational evidence for such rare events like volcanoes, and we now mention that the limited available sampling calls for caution when considering observations only (L48-L50).

Also, I would like to see the CMIP5 Hovmoller diagram next to the observed one. Please make a panel (c) in Figure 1, that is the same as panel (b), but for CMIP5. The authors mention in one small sentence at the end of the paper that the models and observations do not agree on the timing of the El Niño after the eruptions. This is not illustrated and discussed and leaves the impression that the authors want to sweep this under the rug.

This is important, because it highlights the challenge of comparing the sparse observations with the many (and seemingly robust) model results. I do not think anyone expects this paper to solve all the problems around this topic (exactly because it is so hard to prove in observations), so an open discussion of the caveats will benefit the reader.

- Along the same lines as above, in Figure 2 the authors should provide a third row with the same results from observations. Since that figure deals with Pinatubo, there should be sufficient observations available (at least that is stated as the authors' motivation to focus on Pinatubo).

The use of the IPSL-CMR5B model that produces a better seasonality of ENSO than IPS-CM5A, and the new experimental strategy that samples the influence of oceanic initial conditions better allows an improved discussion of this aspect, at lines 88-95 (see answer to reviewer 1 who requested this new model version and experimental strategy). We also made a new figure S4a that shows each eruption year's percentage of El Niño occurrence in the CMIP5 models (as compared to observations in S4b). A summary of this discussion follows. In observations, the Pacific warming tends to peak at the end of the year of the eruption: this is the case in our model as well (new figure 2). When the eruption occurs in presence of an El Niño-favourable state (as in 3 of the 5 observed instances, see new figure S7), the Pacific warming peaks at the end of the year in our model, as in observations (new figure 2b). The large CMIP5 ensemble samples many different initial conditions, and – overall – is representative of neutral condition, and peaks as in our IPSL CM5B model. It is hence possible that the differences between observations and the model are due to a bias towards recharged oceanic states (El Niño

-favourable) in observations. We now mention this in the main text, but also mention model error and internal noise as possible causes for differences, given the small-observed sample.

- There is a number of studies and model datasets available that could help to resolve a bit more the potential convolution of volcanic eruptions during which El Nino is already underway. Froelicher et al. (2013, GBC) ran a very similar setup as the coupled sensitivity experiments presented here with a version of the NCAR model, also targeting Pinatubo. The focus of that paper is on the carbon cycle, so maybe their model results can be revisited with respect to the issue here. More recently, *Douville et al. (2015, GRL) and Lehner et al. (2016, GRL) used so-called POGA experiments with CNRM (Douville) and CESM and GFDL (Lehner). These experiments are fully coupled, but the eastern tropical SSTs are nudged towards observations, so the ENSO evolution is as observed.* Contrasting these simulations with the historical runs of the respective model (which the authors already use here in their CMIP5 analysis) could inform how much of the observed response (e.g., Figure 1 here) is due to the El Nino-volcano coincidence. In fact, that is the main pitch of the Lehner et al. paper, at least with respect to global temperature.

Using coupled models nudged with SSTs will not help identify the processes at work as the simulations would include the SSTs response without allowing identification of the processes leading to such response. The figure below however shows long experiments we ran for another project. This 30-member experiment uses Sato et al (1993; 2012 update version) volcanic Aerosol Optical Depth forcing dataset, but has prescribed wind stress climatology in the ocean component of the model within the tropical Pacific, while the coupling remains unchanged elsewhere. The Bjerknes feedback is effectively switched off in this experiment, and wind stress anomalies over the Pacific can only arise from external forcing such as volcanism. As we can see in Fig R2 below, this experiment reveals the instalment of the wind response over Western equatorial Pacific after both El Chichón and Pinatubo eruptions (see Fig R2). This experiment hence also suggests that volcanic eruptions can induce westerly wind anomalies over the Pacific Ocean and that the mechanism we identify as triggering the initial westerly wind in our two-tier experiments applies also to the El Chichón eruption. Because of length constraints, we decided not to include the results below in the revised paper, but we hope they will convince the reviewer of the robustness of our results.

Fig.R2 Longitude–time section of 5°S–5°N anomalous zonal wind stress (10^{-2} N m^{-2}) over the Pacific area (120°E–80°W) in IPSL-CM5A from 1982 to 1993 in **a** W-Clim 20 members ensemble and **b** ERA-I re-analysis. Contours indicate anomalies significant at the 90% confidence level, based on two-tailed Student *t*-test. In panel **a**, the only significant large-scale westerly wind anomaly in the western Pacific are in fall of the eruption year following the El Chichón (April 1982) and Pinatubo (June 1991) eruptions.

- I like the idea of relative SSTs telling us more than raw SST anomalies about the dynamic coupling of atmosphere and ocean, but the notion that they are co-located with precipitation anomalies does not seem to hold in IPSL (Figure 2b and 2c). Again, observations would be helpful to put the modelling results into perspective. If there is a model bias in IPSL regarding the precipitation response to radiative or dynamic forcing in the tropical Pacific, then I would expect that bias to impact the instalment of the wind response, that is claimed to be the main trigger for the El Nino after volcanoes (as suggested by the LAND experiment). Why did the authors choose to do these

atmosphere-only experiments with IPSL and not CNRM, which has more of a CMIP5-type coupling of relative SSTs and precipitation (see Figure S2)?

The two other reviewers also pointed out that the ENSO timing was problematic in the IPSL-CM5A model, which prompted us to redo the whole set of experiments with the IPSL-CM5B model, which has a much better seasonality (Bellenger et al., 2014): see answer to reviewer 1 for more details on the revised set of experiments. Also running the experiments with the CNRM model would however have involved heavy technical developments and computing resources, which were not available at CERFACS (where CNRM model is run) at this time. We hope that the robustness of our results across CMIP5 models and in the new set of IPSL-CM5B experiments will convince the reviewer.

- The role of the linear ocean model is not explained sufficiently. Since the authors prescribe SSTs in the atmosphere-only experiment “OCEAN”, what does the SSTs derived from the linear model tell us?

We have improved the description of this experimental setup in the main text (L119-L135) and in the supplementary information. In the OCEAN experiment, the wind variability includes all variability that is forced by SST gradients, i.e. both the Bjerknes feedback and the gradients induced by the volcanic radiative forcing, but excludes the response of continental surfaces to the volcanic radiative forcing. The SST in the linear experiment forced by these winds hence also excludes this process.

Minor comments (no particular order):

- There are no numbers given on the relative likelihood of El Niño after volcanoes. How much more likely are they according to this new definition of El Niño with the relative SST anomalies? See for example Maher et al. (2015, GRL).

Figure 1 shows the anomaly of the percentage of chance of an El Niño after a volcanic eruption in the CMIP5 database, computed from RSST (Figure S4a shows a zoom after the historical eruptions and Figure S5 shows it computed from both RSST and SST). Figure S5b shows that the increase in El Niño occurrence rate is not clearly detected using SST instead of RSST (now mentioned in the text). Figure 1 shows a 30 to 60% increase of the occurrence rate of El Niños after major eruptions.

- L91: maybe say “relationship” rather than “cycle”.

The corresponding sentence has been rephrased.

- A few things about Figure 3a: why do the sensitivity experiments with Pinatubo (pink) start systematically higher than the control (black)? Also, the reverse seems to be the case in Figure 3c. Is this due to the sampling from the historical simulations? At least in Figure 3a, taking into account this offset would bring the pink and black lines into agreement, meaning the volcanic forcing has no discernible effect on those cases. Please clarify. Also, are the authors worried about differences in the response due to

sampling from different background climates if they sample across the whole range of the historical simulations? Why is the confidence interval narrower for panel (c)? Is it because there are more samples for ENSO neutral? Please clarify. Along those lines, it would be good to also show the full range, not just the 90% interval, because then the authors could add the observations in Figure 3, allowing for an apples-to-apples comparison. The lines seem quite smooth; was there a filtering applied?

This issue has been corrected in our new experimental protocol, in which all the ensemble members (generated by applying small perturbations on initial SST) start from single oceanic states (representing conditions favourable to El Niño, neutral, La Niña states) just before the eruption: see the answer to reviewer 1 for more details.

- L100: delete “rather“.

Done.

- A suggestion: reorder text and figures so that current Figure 3 comes before Figure 2; currently, the text jumps back and forth between those two figures and it is unnecessarily hard to follow.

Done

- L110: what is meant exactly with “overall response”?

The corresponding sentence has been rephrased.

- Paragraph starting L112: consider providing at least a bit more details on the atmosphere-only experiments here; currently it is not comprehensible what you did. The supplementary is good. Consider adding a simple table along these lines:

Experiment name	Origin of ocean forcing	Origin of land forcing	Atmosphere forcing
Control	Control	Control	No volcanoes
ATM	Control	Control	Volcanoes
LAND	Control	Forced	No volcanoes
OCEAN	Forced	Control	No volcanoes
ALL	Forced	Forced	Volcanoes

Thanks for the suggestion. We now include this table as Supplementary Table S2 and provide detailed information on the method in Section 3.2 (see response to reviewer 1).

- L134: “...stage that is prescribed via the SST anomalies...”
- L139: “the western Pacific”
- L210-212: please clean up the notation of the experiments; at one point the authors talk about “CGCM control ensemble”, at one point about “IPSL-CM5A ensemble”.

All minor changes above have been implemented.

As mentioned earlier, the paper needs to discuss in more detail the implications of the fact that the main sensitivity experiments are based on just one model.

We have added a short discussion on this aspect and mention the upcoming VOLMIP exercise that will provide the modelling inter comparison framework to explore the robustness of our results.

- Figure 4: please add the results from the fully-coupled simulations as well, both for the time series and the maps. Also, add a series of maps for the observations.

We have completely revised figures 3 and 4. These figures are already dense, and the comparison to coupled simulation SST is provided in SI figures S10 (maps) and S11 (time series).

- As mentioned earlier, Figure 4b is a bit confusing, since the SST anomaly in the OCEAN experiment is not the same as in the fully-coupled experiment, even though that is where the SST anomalies have been taken from to prescribe them to the atmosphere-only model setup. Essentially the same is valid in Figure S4b, where there is a clear discrepancy between the ALL results and the OAGCM early on in the simulation.

In the new corresponding Figures all the results are consistent.

REVIEWERS' COMMENTS:

Reviewer #1 (Remarks to the Author):

The authors have invested a remarkable amount of effort in order to address mine and the other reviewers' concerns and the paper is with no-doubt improved. The results of this study will be of great interest to a broad range of scientists and fit well with the scope of the journal. I have only few additional suggestions that should be relatively straightforward for the authors to address.

Additional analysis:

Given that now the main cause of the El Niño-like response is tied to changes over Northern Africa and the West African Monsoon strength. I would suggest showing the changes in Walker circulation as well as the surface ocean conditions (SST) over the equatorial Atlantic as well. Atlantic Niño (Nina) that peaks in summer (JASO) favors La Niña (El Niño) conditions in the following winter (Rodríguez-Fonseca et al. 2009, Li et al. 2016), due to changes in the Walker circulation. The changes in the strength of the WAM likely impact the phase of ATL Niño and consequently may affect ENSO phase. It would therefore be very useful to plot the changes in Walker circulation as well as the longitude–time section of 3°S–3°N anomalous SST (or relative SST) for the Atlantic.

From figure 3 it looks like that the westerly anomalies that develop in summer due to the volcanic forcing (3d) excite a Kelvin wave that travels eastward and reaches the Eastern side of the basin by winter (3e and 3f). In the Neutral case El Niño-like anomaly – albeit weak – already develops in the winter right after the eruption. I wonder why the thermocline anomalies do not manage to reach the eastern Pacific in the La Niña and El Niño case (Figure S9). This should be discussed in the text. Furthermore, figure S9 should be in the main text (adding the panel related to the ocean current as in Figure 3e).

I would suggest extending the methodology part of the main manuscript. Nature communications allows for up to 5000 words and 10 display items. Therefore, some figures from the supplementary can also be placed in the main manuscript as for example the one suggested above. This would avoid the reader referring too often to the supplementary.

Figures:

Figure 2: I suggest adding another set of panels on the side in which you just show the net ENSO-like response: volcano – no-volcano ensemble.

Figure S4 and S7 please add the color legend (for the eruptions) in the panels as in Figure 1

Edits:

LL85-86 please fix it: add “eruption(s)” or “forcing” after “volcanic”. Furthermore, the forcing always triggers El Niño-like “warming” (warming is redundant, since El Niño means already positive temperature anomaly). From figure 2 it look like the volcanic forcing leads to an El Niño.

L95 The authors refer to Pausata et al. (2016), which is fine; however, the first study suggesting the southward displacement of ITCZ movement as cause of El Niño-like response due to the NH cooling is Pausata et al. (2015).

L145 Odd way of referencing: “The reduced precipitation and tropospheric heating²⁰ in the equatorial latitudes drive ...” why the reference there?

References:

Li, X., Xie, S.-P., Gille, S. T. & Yoo, C. Atlantic-induced pan-tropical climate change over the past three decades. *Nat. Clim. Chang.* 6, 275–279 (2016).

Pausata, F. S. R., L. Chafik, R. Caballero, and D. S. Battisti (2015), Impacts of high-latitude volcanic eruptions on ENSO and AMOC, *Proc. Natl. Acad. Sci.*, 201509153, doi:10.1073/pnas.1509153112.

Pausata, F. S. R., Karamperidou, C., Caballero, R., and D. S. Battisti. ENSO response to high-latitude volcanic eruptions in the Northern Hemisphere: the role of initial conditions, *Geophys. Res. Lett.*, 43, 8694–8702 (2016).

Rodríguez-Fonseca, B. et al. Are Atlantic Niños enhancing Pacific ENSO events in recent decades? *Geophys. Res. Lett.* 36, L20705 (2009).

Reviewer #2 (Remarks to the Author):

A. Synopsis/summary of key results

This is a revised version of a study revisiting ENSO response to strong tropical volcanic eruptions. During the revision, the authors have updated their experimental design, in particular using a coupled climate model that better represents some key ENSO features and expanding the set of sensitivity experiments to better identify the key dynamics leading to the post-eruption El Niño event. The revision is substantial. Some of the conclusions in the original study are confirmed or even corroborated in this new version (for instance, the El Niño response is found to be rather insensitive to the preconditioning). The major difference in the conclusions is that the authors identify now the cooling over tropical Africa as the main source for the initial ENSO anomaly, while it was the land-sea contrast in their previous assessment.

B. Originality and significance

The topic of how ENSO responds to volcanic forcing remains relevant and unsettled. Since the original submission, only a few new papers have been published on the topic. They do not affect the overall originality and significance of this contribution, particularly concerning the dynamical interpretation of the ENSO response.

C.- F. Data & methodology, statistics and uncertainties, Conclusions, Suggested improvements

I had a few minor concerns and requests for clarification about the original submission, which were overall satisfactorily accounted for by the authors.

The various methodological aspects of the revised study appear more robust and cleaner compared to the original version. In particular the explanation attributing a major role to the cooling over tropical Africa appears much more convincing and better substantiated than the volcanic breeze hypothesis proposed in the original study.

On the conclusion about necessity to include volcanic forcing in operational seasonal forecast: I think this is not straightforward and requires careful representation of the forcing. As also mentioned in this paper, volcanic forcing produces a stronger than observed tropical cooling, therefore introducing a potential bias which could be detrimental for observational forecasts, and

hamper the advantages of including a representation of volcanically-forced ENSO responses.

Figure S5: panel (a) looks odd as having a 70% chance of occurrence of an El Nino every year suggests a methodological flaw. Was the long-term trend of the index subtracted before calculating the standard deviation (I think it should be removed)?

G: References

References are properly accounted for, including very recent papers.

H. Clarity and context

The writing and the presentation of the results are clear, and the paper is well structured.

Some editorial minors are listed below:

Abstract: isn't the abstract in Nat Comm unreferenced?

Line 43: it is the NINO3.4 index based on SST, not on RSST, that is a common ENSO indicator

Line 83: even the RSST anomalies are very weak in the CMIP5 ensemble shown in Fig S3 with rarely 2/3 of the ensemble having same sign. Is this really a strong enough signal to justify this statement?

Line 103: shown is wind stress

Line 107: check "surface a"

Line 157: from "the" equator

Line 163: please define "normal", you mean non-volcanic? Again there are many flavors or types of El Nino events, so better specify to avoid confusion

Line 191: "VolMIP"

Fig. 3 panel d: shown is wind stress (same for Fig S6)

Fig. 3 panel e: specify eastward is positive?

Fig.3 panels g,h: shading is the confidence band for the ALL experiment only

Fig. 4: "at the end of the eruption year"

Fig. 4, last sentence: "at the 90% confidence ..."

Supp Fig S2: check format of line numbering

Reviewer #3 (Remarks to the Author):

I congratulate the authors for this thorough revision. The significant effort to rerun the experimental setup with a more realistic model is commendable. I think it was worth it, since the study is more convincing now. The authors have also provided comprehensive additional material and explanation in response to the review comments. At this point, I only have a few minor

additional comments that might be taken into account before publication.

Figure S14 and S15: while the LAND experiment nicely reproduces temperature and precipitation anomalies of the OAGCM experiments, it seems to have too strong of a SLP response in Asia and Australia compared to OAGCM. Maybe the authors can discuss this briefly. Generally, the experimental setup is very thorough, but I did miss a discussion of potential non-linearities, such as the strong SLP response in LAND that appears to be damped somehow in OAGCM. Along those lines, it would be interesting to simply add up the different sensitivity experiments to see to what extent linearity holds.; for example, Figs. S14 and S15, but also Fig. 3g-h. Or the authors could add the OAGCM results to Fig. 3g-h to compare with ALL.

L75: the word "twin" by itself does not allow to understand what has been done. I'm not a native speaker myself, but the readability of the text in some places suffers from the authors trying to cram too much information into the text (I know, word limit...). I recommend avoiding the word "twin" in favor of "pair", since that's used regularly later in the paper, and adding a sentence. Just a suggestion: "[...] we ran three *pairs* of 30-member ensembles of length 3 years, starting from [...]. The *pairs* consists of 30 member ensembles with and without volcanic eruption. In the absence of [...]"

L107: delete "a"

L177, sentence starting with "Our...": suggests to write "mechanism different" to make clear that the end of the sentence is what the authors here propose and not what references 10,16 propose.

The authors do a relative poor job of putting their results into perspective of the existing literature, which I think is important to convince other scientists that the new explanation is superior to existing ones. Is there a way to reconcile some of the suggested mechanisms? For example (I'm just speculating here), is there also an ITCZ shift occurring in the IPSL model (ref16) or are some of the mechanisms proposed by ref10 recovered in the IPSL model, despite the "mistake" of ref10 to focus on SST rather than RSST?

After the conservative final statements of the authors I'm left wondering what the implications for CMIP5 are? The authors showed in response to the reviews that in observations there is very little difference between SST and RSST (Fig. 1 and Fig. S1), but show that there is a large difference in CMIP5 (Fig. S3). The authors also showed that in IPSL it does not matter that much in which initial state the model is before the eruption, it will always tend to produce an El Nino response. In fact, for neutral conditions the effect of the eruption on El Nino odds is probably highest (Fig. 2). Since CMIP5 models, on average, are in neutral conditions before an eruption, the effect of the eruption should actually be visible in CMIP5, both in SST and RSST. So how can we explain this discrepancy? As mentioned in the first round of reviews, I don't think this can be blamed on a systematic overestimation of the post-eruption cooling in CMIP5, since that is an artifact of the model sampling (Lehner et al., 2016, GRL). Maybe it is beyond the scope of the study to speculate about possible reasons. Also, VOLMIP will hopefully help to solve some of these questions. Personally, however, I would be interested in knowing the authors opinion on the CMIP5 difference in SST and RSST, since that is an important take-away from this paper.

Despite the convincing arguments, I would change the title to "How tropical explosive volcanic eruptions *can* trigger El Nino", since this is a single model study and we have to wait for VOLMIP to make things more robust, and also since we know El Nino is not *always* triggered after an eruption.

Reviewer #1 (Remarks to the Author):

The authors have invested a remarkable amount of effort in order to address mine and the other reviewers' concerns and the paper is with no-doubt improved. The results of this study will be of great interest to a broad range of scientists and fit well with the scope of the journal. I have only few additional suggestions that should be relatively straightforward for the authors to address.

We thank the reviewer for his positive feedbacks on the revised version of his manuscript, and for all his comments, which allowed us to significantly improve the manuscript.

Additional analysis:

Given that now the main cause of the El Niño-like response is tied to changes over Northern Africa and the West African Monsoon strength. I would suggest showing the changes in Walker circulation as well as the surface ocean conditions (SST) over the equatorial Atlantic as well. Atlantic Niño (Nina) that peaks in summer (JASO) favors La Niña (El Niño) conditions in the following winter (Rodríguez-Fonseca et al. 2009, Li et al. 2016), due to changes in the Walker circulation. The changes in the strength of the WAM likely impact the phase of ATL Niño and consequently may affect ENSO phase. It would therefore be very useful to plot the changes in Walker circulation as well as the longitude–time section of 3°S–3°N anomalous SST (or relative SST) for the Atlantic.

There is indeed an Atlantic Niño developing in Summer-Fall of the eruption year as a response to the reduced West African Monsoon strength (which induces westerly winds over the Atlantic through atmospheric Rossby waves). We have modified the Figure 7 (former Figure 4) of the main manuscript to show the equatorial Atlantic response. We now mention this in the main text and cite both suggested references. See lines 222-231.

From figure 3 it looks like that the westerly anomalies that develop in summer due to the volcanic forcing (3d) excite a Kelvin wave that travels eastward and reaches the Eastern side of the basin by winter (3e and 3f). In the Neutral case El Niño-like anomaly – albeit weak – already develops in the winter right after the eruption. I wonder why the thermocline anomalies do not manage to reach the eastern Pacific in the La Niña and El Niño case (Figure S9). This should be discussed in the text. Furthermore, figure S9 should be in the main text (adding the panel related to the ocean current as in Figure 3e).

The statistically-significant differences in the thermocline depth signals induced by the volcanic forcing only concern a couple of points very close to the coast in the eastern equatorial Pacific. The 90% confidence masking on figure 3 implies that 10% of the points marked as significant on the figure may not be significant: this may be the reason why this local difference appears close to the South American coast. Except from these few points, the thermocline signal is very consistent between the three experiments.

We decided not to include Figure S9 in the main text, as it is redundant with the new figure 2. Regarding the suggestion of adding the related ocean current on Figure S9, this was not possible since we didn't save all 3D ocean variables for the La Niña and El Niño ensemble due to storage issues.

I would suggest extending the methodology part of the main manuscript. Nature communications allows for up to 5000 words and 10 display items. Therefore, some figures from the supplementary can also be placed in the main manuscript as for example the one suggested above. This would avoid the reader referring too often to the supplementary.

Thanks for the suggestion. We now added 2 main figures: Former Supplementary Figure 10 is now the new Figure 4. Former Supplementary Figure 12 is now part of the new Figure 5 and former Supplementary Figure 13 is now the main Figure 6. We felt that the rest of the figures and methodology belonged to the supplementary information, in order to keep the flow of the main text.

Figures:

Figure 2: I suggest adding another set of panels on the side in which you just show the net ENSO-like response: volcano – no-volcano ensemble.

Thanks for the suggestion. Figure 2 has been modified accordingly.

Figure S4 and S7 please add the color legend (for the eruptions) in the panels as in Figure 1

Thanks for the suggestion. Figure S4 and S7 have been modified accordingly.

Edits:

LL85-86 please fix it: add “eruption(s)” or “forcing” after “volcanic”.

Done

Furthermore, the forcing always triggers El Nino-like “warming” (warming is redundant, since El Nino means already positive temperature anomaly). From figure 2 it look like the volcanic forcing leads to an El Nino.

Sorry, but we did not understand this comment.

L95 The authors refer to Pausata et al. (2016), which is fine; however, the first study suggesting the southward displacement of ITCZ movement as cause of El Nino-like response due to the NH cooling is Pausata et al. (2015).

We now cite both references (now number references 21 and 22).

L145 Odd way of referencing: “The reduced precipitation and tropospheric heating²⁰ in the equatorial latitudes drive ...” why the reference there?

We have moved the reference to after mentioning the “Matsuno-Gill” response.

Reviewer #2 (Remarks to the Author):

A. Synopsis/summary of key results

This is a revised version of a study revisiting ENSO response to strong tropical volcanic eruptions. During the revision, the authors have updated their experimental design, in particular using a coupled climate model that better represents some key ENSO features and expanding the set of sensitivity experiments to better identify the key dynamics leading to the post-eruption El Niño event. The revision is substantial. Some of the conclusions in the original study are confirmed or even corroborated in this new version (for instance, the El Niño response is found to be rather insensitive to the preconditioning). The major difference in the conclusions is that the authors identify now the cooling over tropical Africa as the main source for the initial ENSO anomaly, while it was the land-sea contrast in their previous assessment.

We thank the reviewer for his positive feedbacks on the revised version of his manuscript, and for all his comments, which allowed us to significantly improve the manuscript.

B. Originality and significance

The topic of how ENSO responds to volcanic forcing remains relevant and unsettled. Since the original submission, only a few new papers have been published on the topic. They do not affect the overall originality and significance of this contribution, particularly concerning the dynamical interpretation of the ENSO response.

C.- F. Data & methodology, statistics and uncertainties, Conclusions, Suggested improvements
I had a few minor concerns and requests for clarification about the original submission, which were overall satisfactorily accounted for by the authors.

The various methodological aspects of the revised study appear more robust and cleaner compared to the original version. In particular the explanation attributing a major role to the cooling over tropical Africa appears much more convincing and better substantiated than the volcanic breeze hypothesis proposed in the original study.

On the conclusion about necessity to include volcanic forcing in operational seasonal forecast: I think this is not straightforward and requires careful representation of the forcing. As also mentioned in this paper, volcanic forcing produces a stronger than observed tropical cooling, therefore introducing a potential bias which could be detrimental for observational forecasts, and hamper the advantages of including a representation of volcanically-forced ENSO responses.

We agree with the reviewer. We modified the corresponding sentence to emphasize that a better representation of the response to volcanic forcing in climate models could open the path to improved ENSO predictability after eruptions. See line 303-305.

Figure S5: panel (a) looks odd as having a 70% chance of occurrence of an El Niño every year suggests a methodological flaw. Was the long-term trend of the index subtracted before calculating the standard deviation (I think it should be removed)?

Figure S5a (based on raw SST) is only meant to be compared with figure S5b (based on relative SST): while the ENSO response to volcanic eruptions is clear when using relative SST, it is less

straightforward to interpret when using absolute SST, due to the global cooling signal after eruptions and global warming signal. Figure S5 caption has been improved to convey that message better.

G: References

References are properly accounted for, including very recent papers.

H. Clarity and context

The writing and the presentation of the results are clear, and the paper is well structured.

Some editorial minors are listed below:

Abstract: isn't the abstract in Nat Comm unreferenced?

We have revised the abstract accordingly.

Line 43: it is the NINO3.4 index based on SST, not on RSST, that is a common ENSO indicator. The classical Niño index is indeed based on SST, not RSST. But the analysis of figure S5 clearly shows that using RSST allows to better detect the ENSO response to volcanism in climate models.

Line 83: even the RSST anomalies are very weak in the CMIP5 ensemble shown in Fig S3 with rarely 2/3 of the ensemble having same sign. Is this really a strong enough signal to justify this statement?

Figure S5 shows that volcanic forcing only slightly increases the chances for an El Niño (from ~30% to 40-50%). As a result, the ensemble average of figure S3 will be biased towards EL Niño conditions, but will only be a fraction of a typical El Niño signal. We have changed "induces" by "tends to favour" an El Niño to convey that.

Line 103: shown is wind stress

Correct but these changes are consistent with the large-scale wind changes. The new figure 4 (which shows wind) is now also referenced.

Line 107: check "surface a"

Thanks, this has been corrected.

Line 157: from "the" equator

Thanks, this has been corrected.

Line 163: please define "normal", you mean non-volcanic? Again there are many flavors or types of El Niño events, so better specify to avoid confusion

We have rephrased to "internally generated El Niño".

Line 191: "VolMIP"

Thanks, this has been corrected.

Fig. 3 panel d: shown is wind stress (same for Fig S6)

Thanks, this has been corrected.

Fig. 3 panel e: specify eastward is positive?

We added this sentence in the figure caption: “On panel e eastward current corresponds to positive anomalies.”

Fig.3 panels g,h: shading is the confidence band for the ALL experiment only

We added this sentence in the figure caption of the new Figure 5: “The shading indicates the 90% confidence interval for the ALL experiment.”

Fig. 4: “at the end of the eruption year”

Thanks, this has been corrected.

Fig. 4, last sentence: “at the 90% confidence ...”

Thanks, this has been corrected.

Supp Fig S2: check format of line numbering

This was a “Word to PDF” conversion issue. Solved.

Reviewer #3 (Remarks to the Author):

I congratulate the authors for this thorough revision. The significant effort to rerun the experimental setup with a more realistic model is commendable. I think it was worth it, since the study is more convincing now. The authors have also provided comprehensive additional material and explanation in response to the review comments. At this point, I only have a few minor additional comments that might be taken into account before publication.

We thank the reviewer for his positive feedbacks on the revised version of his manuscript, and for all his comments, which allowed us to significantly improve the manuscript.

Figure S14 and S15: while the LAND experiment nicely reproduces temperature and precipitation anomalies of the OAGCM experiments, it seems to have too strong of a SLP response in Asia and Australia compared to OAGCM. Maybe the authors can discuss this briefly. Generally, the experimental setup is very thorough, but I did miss a discussion of potential non-linearities, such as the strong SLP response in LAND that appears to be damped somehow in OAGCM. Along those lines, it would be interesting to simply add up the different sensitivity experiments to see to what extent linearity holds.; for example, Figs. S14 and S15, but also Fig. 3g-h. Or the authors could add the OAGCM results to Fig. 3g-h to compare with ALL.

The linearity holds for precipitation and surface temperature anomalies. This is now illustrated for surface temperature by supplementary figure 10, which shows that the LAND+OCEAN (the ATM response being negligible) is quite similar to the ALL and OAGCM responses. We enclose the same comparison for SLP changes in fall of the eruption year below. There are indeed differences in amplitudes in some regions (maybe due to saturation effects), but the broad SLP (and most importantly for our study: wind) patterns generally agree.

L75: the word “twin” by itself does not allow to understand what has been done. I’m not a native speaker myself, but the readability of the text in some places suffers from the authors trying to cram too much information into the text (I know, word limit...). I recommend avoiding the word “twin” in favor of “pair”, since that’s used regularly later in the paper, and adding a

sentence. Just a suggestion: “[...] we ran three *pairs* of 30-member ensembles of length 3 years, starting from [...]. The *pairs* consists of 30 member ensembles with and without volcanic eruption. In the absence of [...]”

Done. See line 140-143.

L107: delete “a”

Done

L177, sentence starting with “Our...”: suggests to write “mechanism different” to make clear that the end of the sentence is what the authors here propose and not what references 10,16 propose.

Done

The authors do a relative poor job of putting their results into perspective of the existing literature, which I think is important to convince other scientists that the new explanation is superior to existing ones. Is there a way to reconcile some of the suggested mechanisms? For example (I’m just speculating here), is there also an ITCZ shift occurring in the IPSL model (ref16) or are some of the mechanisms proposed by ref10 recovered in the IPSL model, despite the “mistake” of ref10 to focus on SST rather than RSST?

We now discuss this in more details when presenting the results (new Figures 4 and 5 and lines 267-283).

After the conservative final statements of the authors I’m left wondering what the implications for CMIP5 are? The authors showed in response to the reviews that in observations there is very little difference between SST and RSST (Fig. 1 and Fig. S1), but show that there is a large difference in CMIP5 (Fig. S3). The authors also showed that in IPSL it does not matter that much in which initial state the model is before the eruption, it will always tend to produce an El Nino response. In fact, for neutral conditions the effect of the eruption on El Nino odds is probably highest (Fig. 2). Since CMIP5 models, on average, are in neutral conditions before an eruption, the effect of the eruption should actually be visible in CMIP5, both in SST and RSST. So how can we explain this discrepancy? As mentioned in the first round of reviews, I don’t think this can be blamed on an systematic overestimation of the post-eruption cooling in CMIP5, since that is an artifact of the model sampling (Lehner et al., 2016, GRL). Maybe it is beyond the scope of the study to speculate about possible reasons. Also, VOLMIP will hopefully help to solve some of these questions. Personally, however, I would be interested in knowing the authors opinion on the CMIP5 difference in SST and RSST, since that is an important take-away from this paper.

We now discuss this in the “Discussion” section. See lines 259-263.

Despite the convincing arguments, I would change the title to “How tropical explosive volcanic eruptions *can* trigger El Nino”, since this is a single model study and we have to wait for VOLMIP to make things more robust, and also since we know El Nino is not *always* triggered after an eruption.

We modified the title as suggested.